# Restoring the top-of-atmosphere reflectance during solar eclipses: a proof of concept with the UV Absorbing Aerosol Index measured by TROPOMI

Victor Trees[1,2], Ping Wang[1], and Piet Stammes[1]

[1]Royal Netherlands Meteorological Institute (KNMI), De Bilt, The Netherlands
[2]Delft University of Technology, Delft, The Netherlands

**Correspondence:** Victor Trees (victor.trees@knmi.nl)

**Abstract.** During a solar eclipse the solar irradiance reaching the top-of-atmosphere (TOA) is reduced in the Moon shadow. The solar irradiance is commonly measured by Earth observation satellites before the start of the solar eclipse and is not corrected for this reduction, which results in a decrease of the computed TOA reflectances. Consequently, air quality products that are derived from TOA reflectance spectra, such as the ultraviolet (UV) Absorbing Aerosol Index (AAI), are distorted or undefined in the shadow of the Moon. The availability of air quality satellite data in the penumbral and antumbral shadow during solar eclipses, however, is of particular interest to users studying the atmospheric response to solar eclipses. Given the time and location of a point on the Earth's surface, we explain how to compute the obscuration during a solar eclipse taking into account wavelength-dependent solar limb darkening. With the calculated obscuration fractions, we restore the TOA reflectances and the AAI in the penumbral shadow during the annular solar eclipses on 26 December 2019 and 21 June 2020 measured by the TROPOMI/S5P instrument. We compare the calculated obscuration to the estimated obscuration using an uneclipsed orbit. In the corrected products, the signature of the Moon shadow disappeared, but only if wavelength-dependent solar limb darkening is taken into account. We find that the Moon shadow anomaly in the uncorrected AAI is caused by a reduction of the measured reflectance at 380 nm, rather than a color change of the measured light. We restore common AAI features such as the sunglint and desert dust, and we confirm the restored AAI feature on 21 June 2020 at the Taklamakan desert by measurements of the GOME-2C satellite instrument on the same day but outside the Moon shadow. No indication of local absorbing aerosol changes caused by the eclipses was found. We conclude that the correction method of this paper can be used to detect real AAI rising phenomena during a solar eclipse and has the potential to restore any other product that is derived from TOA reflectance spectra. This would resolve the solar eclipse anomalies in satellite air quality measurements in the penumbra and antumbra, and would allow for studying the effect of the eclipse obscuration on the composition of the Earth's atmosphere from space.

## 1 Introduction

Earth observation satellite spectrometer instruments are designed to measure the particles and gases in the Earth's atmosphere. They rely upon the reflectance of the incident sunlight on the top-of-atmosphere (TOA) at various wavelengths in the UV, visible, near-infrared and shortwave-infrared spectral domains. These TOA reflectances are calculated through the division

of the measured Earth radiance by the measured solar irradiance. During a solar eclipse, the solar irradiance reaching TOA

is reduced as the Moon blocks (part of) the sunlight, reducing the Earth radiance. Because the solar irradiance is commonly measured before the start of the eclipse, the atmosphere measurements are distorted in the shadow of the Moon or, after raising an eclipse flag, undefined.

Since the start of the nominal operational mode of the TROPOMI spectrometer instrument on board the S5P satellite in May 2018, 7 solar eclipses occurred, 6 of which have been measured by TROPOMI. An example of an air quality product of

TROPOMI that suffers from the Moon shadow is the ultraviolet (UV) Absorbing Aerosol Index (AAI). The AAI is a qualitative measure of absorbing aerosols in the atmosphere such as desert dust, volcanic ash and anthropologically produced soot, and can be used to daily and globally track the aerosol plumes from dust storms, forest fires, volcanic eruptions and biomass burning. The AAI is retrieved from TOA reflectance measurements at two wavelengths in the UV-range, hence the AAI may directly be affected by the obscuration during a solar eclipse. Figure 1 is a near-global AAI map on 21 June 2020, using TOA

reflectance measurements at 340 nm and 380 nm by TROPOMI. Dust aerosol plumes over the Atlantic Ocean originating from the Sahara can be identified through the AAI increase of $\sim 2$ to $\sim 4$ points relative to their surrounding regions. In Western China an AAI larger than 5 is measured, which is caused by the shadow of the Moon. TROPOMI data contains an eclipse flag indicating the eclipse occurrence at a ground pixel. For satellite instruments that do not contain an eclipse flag, such as the GOME-2 instrument, these eclipse anomalies propagate into anomalies in temporal average maps, potentially resulting in false

conclusions about the mean aerosol effect in that time period.[1]

The reduction of the solar irradiance during an eclipse might influence the photochemical activity, and therefore the composition, of the Earth's atmosphere. Measurements of the speed and significance of this atmospheric response could contribute to the understanding of the sensitivity of planetary atmospheres to (variations in) their solar or stellar illumination and could possibly be used to verify atmospheric chemistry models. Ground-based measurements during solar eclipses of local ozone

column fluctuations have been taken using Dobson and Brewer spectrophotometers, but the reported results are contradictory (see e.g. Bojkov, 1968; Mims and Mims, 1993; Chakrabarty et al., 1997, 2001). Zerefos et al. (2000) pointed out the importance of solar limb darkening and the direct to diffuse irradiance on the ozone column retrieval, but also the change in effective temperature in the ozone layer or other atmospheric conditions (different cloudiness, solar zenith angle and turbidity) may have influenced the measurements (Winkler et al., 2001). Unambiguous increases in local $NO_2$ concentration have been measured

from the ground during solar eclipses resulting from the reduced photodissociation of $NO_2$ in the stratosphere (see e.g. Gil et al., 2000; Adams et al., 2010). Unlike ozone, $NO_2$ reacts on a timescale of several minutes directly responding to the eclipse obscuration (Herman, 1979; Wuebbles and Chang, 1979). Although similar information could be obtained during sunrise and sunset, Wuebbles and Chang (1979) pointed out that the relatively short time durations of solar eclipses allow for a more clear identification of the major photochemical cycles in the stratosphere, due to the smaller bias from atmospheric transport, mixing

and interfering chemical reactions throughout the diurnal cycle. Ground-based measurements, however, are taken at a single

---

[1] An example of a *monthly average* AAI map of the GOME-2 satellite instrument that is distorted by a solar eclipse can be found on https://d1qb6yzwaaq4he. cloudfront.net/airpollution/absaai/GOME2B/monthly/images/2019/GOME-2B_AAI_map_201912.png, visited on 22 February 2021.

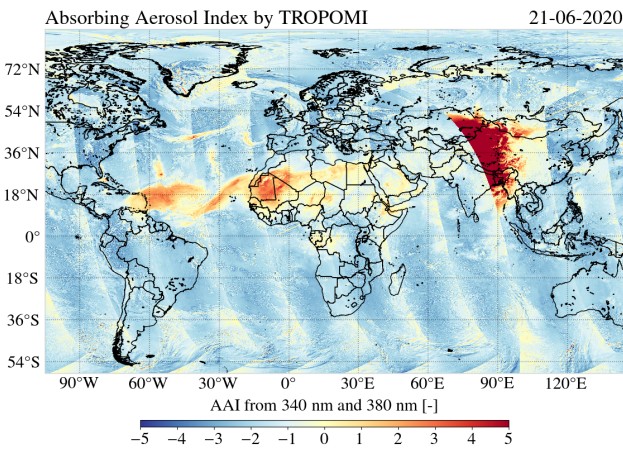

**Figure 1.** The Absorbing Aerosol Index from the 340/380 nm wavelength pair by TROPOMI on 21 June 2020. The anomaly centered at Western China is caused by the shadow of the Moon.

location. Being able to restore satellite data in the Moon shadow would allow for studying the effect of solar eclipses on the Earth's atmosphere from space at various locations with the same instrument.

The geometry of the Moon shadow on the Earth's surface is an astronomical well-understood problem and predictions of the eclipse time, location and local eclipse circumstances can be done with high accuracy (Espenak and Meeus, 2006; Meeus, 1989). The eclipse obscuration at a point in the shadow can be approximated by the fraction of the area of the apparent solar disk occulted by the Moon (Seidelmann, 1992). Montornès et al. (2016) approximated the eclipse obscuration by the fraction of the solar disk *diameter* occulted by the Moon in order to correct the TOA solar irradiance in the Advanced Research Weather and Forecasting (WRF-ARW) model and modeled a local surface temperature response of $\sim -1$ K to $\sim -3$ K, with a time lag between $\sim 5$ and $\sim 15$ minutes after the instant of maximum obscuration. Such wavelength-independent approximations of the eclipse obscuration fraction based on the the overlapping disks indeed could work well to estimate the shortwave fluxes, depending on the desired accuracy. If the spectral variation of the measured light is to be studied, however, the wavelength dependence of the eclipse obscuration fraction, caused by solar limb darkening, cannot be neglected. Koepke et al. (2001) provided the formulae to compute the eclipse obscuration fraction for total eclipses taking into account solar limb darkening if the relative position and apparent dimensions of the lunar and solar disks are known. They showed that the error in the solar irradiance close to total obscuration may become 30% at 1500 nm and 60% at 310 nm if solar limb darkening is not taken into account.

Emde and Mayer (2007), Kazantzidis et al. (2007) and Ockenfuß et al. (2020) performed extensive 3-D radiative transfer modelling of total solar eclipses, taking into account solar limb darkening. Their work gives insight into the spectral behaviour of sunlight reaching a ground sensor located in or close to the total Moon shadow and the importance of the various 3-D radiative transfer components. Emde and Mayer (2007) pointed out that solar eclipses provide excellent opportunities to test

3-D radiative transfer codes against measurements because, unlike broken cloud fields, the Moon shadow's geometry is well-defined.

In this paper, we present a method to restore the TOA reflectance as measured by Earth observation satellites in the penumbra and antumbra of solar eclipses, by combining accurate eclipse predictions with the computation of the eclipse obscuration fraction taking into account wavelength-dependent solar limb darkening. We apply this method to the TOA reflectances measured by the TROPOMI/S5P satellite instrument in the penumbra during the annular solar eclipses on 26 December 2019 and 21 June 2020, and we show how the calculated obscuration fraction can be compared to the estimated obscuration fraction from measurements in an uneclipsed orbit. With the restored TOA reflectances, we compute a corrected version of the AAI and analyze the features that were otherwise hidden in the shadow of the Moon.

This paper is structured as follows. In Sect. 2, we explain the method to restore the measured TOA reflectance during a solar eclipse. In Sect. 3, we show the results of applying this method to the eclipsed TROPOMI orbits during the annular solar eclipses on 26 December 2019 and 21 June 2020. In Sect. 4, we discuss the limits of the method and the points of attention for future applications. In Sect. 5, we summarize the results and state the most important conclusions of this paper.

## 2 Method

Here, we explain the method to restore the measured TOA reflectance during a solar eclipse. We start with explaining the situation of measuring the TOA reflectance during a solar eclipse and how these measurements can be restored with the eclipse obscuration fraction (Sect. 2.1). Then, we explain the Moon shadow types (Sect. 2.2) and how we compute the eclipse obscuration fraction taking into account solar limb darkening, knowing the local eclipse circumstances (Sect. 2.3). In Sect. 2.4 and Appendix A we explain how we compute the local eclipse circumstances from the measurement time and location of a point on the Earth's surface.

### 2.1 Solar irradiance correction

The spectral TOA reflectance of an atmosphere-surface system as measured by a satellite is defined as

$$R^{\text{meas}}(\lambda) = \frac{\pi I(\lambda)}{\mu_0 E_0(\lambda)}, \tag{1}$$

where $I$ is the radiance reflected by the atmosphere-surface system in W m$^{-2}$sr$^{-1}$nm$^{-1}$ and $E_0$ is the extraterrestrial solar irradiance perpendicular to the beam in W m$^{-2}$nm$^{-1}$. The units nm$^{-1}$ indicate that both $I$ and $E_0$ depend on wavelength $\lambda$. Also, $I$ depends on the viewing zenith angle $\theta$, the solar zenith angle $\theta_0$, the viewing azimuth angle $\varphi$ and the solar azimuth angle $\varphi_0$. Furthermore, we use the definitions $\mu = \cos\theta$ and $\mu_0 = \cos\theta_0$. $I$ is measured by TROPOMI continuously at the dayside of the Earth. $E_0$ is measured by TROPOMI near the North Pole once every 15 orbits, which is approximately once every calendar day.

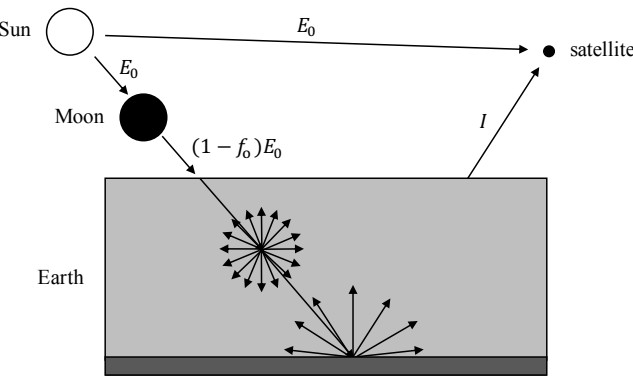

**Figure 2.** Schematic sketch of a satellite retrieving the top-of-atmosphere reflectance during a solar eclipse. $I$ is the measured radiance reflected by the atmosphere-surface system, $f_\mathrm{o}$ is the eclipse obscuration fraction and $E_0$ is the uneclipsed solar irradiance.

During a solar eclipse, the solar irradiance reaching TOA is reduced, as illustrated in Fig. 2. The fraction of the spectral irradiance $E_0(\lambda)$ that is blocked by the Moon is the wavelength-dependent eclipse obscuration fraction, $f_\mathrm{o}(\lambda)$. The remaining spectral irradiance at TOA is $[1\text{-}f_\mathrm{o}(\lambda)]E_0(\lambda)$. We neglect the contribution of the Sun's corona[2]. The intrinsic spectral reflectance of the atmosphere-surface system (i.e. the fraction of the emerging radiance to the incident irradiance), is then obtained by correcting the solar irradiance:

$$R^\mathrm{int}(\lambda) = \frac{\pi I(\lambda)}{\mu_0[1 - f_\mathrm{o}(\lambda)]E_0(\lambda)}. \tag{2}$$

If the optical properties of the atmosphere-surface system are constant just before and during the eclipse, then $R^\mathrm{int}$ is expected to be constant regardless of the eclipse conditions. We compute $R^\mathrm{int}$ from $R^\mathrm{meas}$ by combining Eq. 1 and 2:

$$R^\mathrm{int}(\lambda) = \frac{R^\mathrm{meas}(\lambda)}{1 - f_\mathrm{o}(\lambda)}. \tag{3}$$

Properties of the atmosphere-surface system can be derived during a solar eclipse from the spectrum of $R^\mathrm{int}$. Note that potential changes of the atmosphere-surface system that are caused by the eclipse obscuration may affect $R^\mathrm{int}$, depending on the significance and nature of these changes.

We assume that the solar irradiance is randomly polarized. Also, we neglect light travelling horizontally in the atmosphere from one ground pixel to the other. The importance of horizontal light travelling between adjacent pixels is expected to increase with increasing $f_\mathrm{o}$, but will only become significant close to totality (Emde and Mayer, 2007). In Sect. 4, we reflect back on these assumptions.

---

[2]Emde and Mayer (2007) estimated that the radiance of the corona is approximately $1.7 \cdot 10^{-7}$ times smaller than the radiance originating from the center of solar disk.

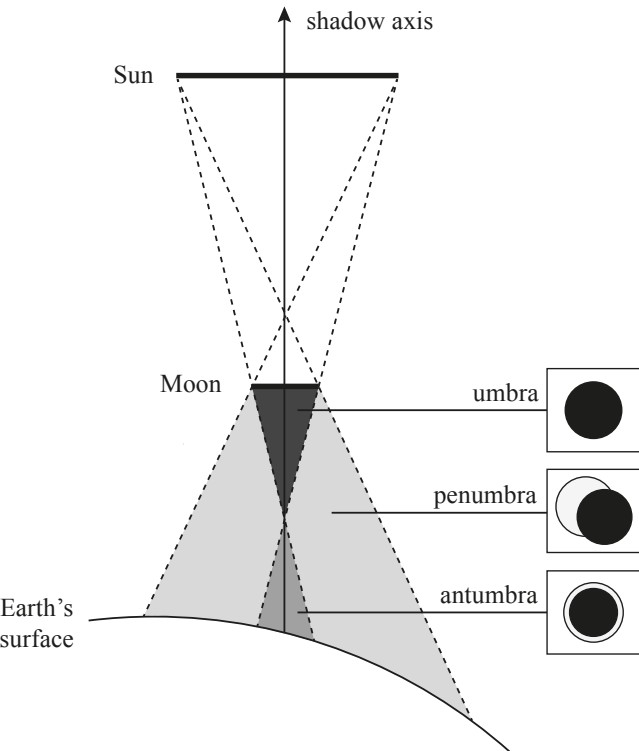

**Figure 3.** Sketch of the Moon shadow types that may occur during a solar eclipse (not to scale). In this example, an annular solar eclipse is experienced at the Earth's surface.

## 2.2 Moon shadow types

The experienced obscuration fraction $f_o(\lambda)$ depends on the location with respect to the position of the Sun and the Moon. Figure 3 illustrates the shadow types that may be experienced during a solar eclipse: (1) the umbra, where the lunar disk fully occults the solar disk ($f_o(\lambda) = 1$) during a total eclipse, (2) the antumbra, where every part of the lunar disk occults the solar disk but full obscuration is not reached ($0 < f_o(\lambda) < 1$) during an annular eclipse, and (3) the penumbra, where only a part of the lunar disk occults the solar disk ($0 < f_o(\lambda) < 1$) during a partial, total or annular eclipse. The Moon-Sun axis is often referred to as the 'shadow axis' as indicated in Fig. 3. The penumbra is always present during an eclipse. Whether an umbra or an antumbra is present on the Earth's surface depends on the distances to the Moon and the Sun, which vary in time as the Moon orbits the Earth and the Earth orbits the Sun, both in elliptical orbits.

In this paper, we do not study the umbra because Eq. 3 breaks down when $f_o(\lambda) = 1$. The solar irradiance correction only applies to pixels located in the penumbra or antumbra, and for which the signal-to-noise is sufficient (we set the constraint $R^{meas} > 50\sigma$ where $\sigma$ is the 1 standard deviation of $R^{meas}$). It is important to note that the area in the penumbra on the Earth's surface is always much larger than the area in the (ant)umbra on the Earth's surface, as will be shown in Sect. 3.

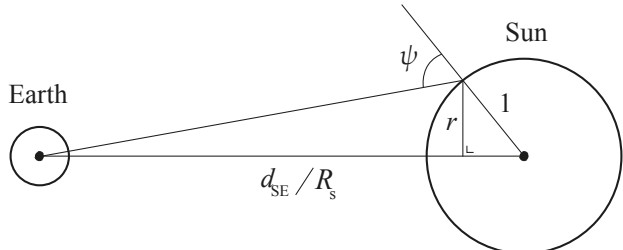

**Figure 4.** Definition of the heliocentric angle $\psi$ used in Eq. 5 (not to scale). Distance $d_{\text{SE}}$ is the geocentric Earth-Sun distance and $r$ is the apparent distance on the solar disk from the solar disk center. All dimensions are per solar radius $R_{\text{s}}$. We neglect the effect of the sphericity of the Sun on the apparent solar disk radius, such that $r = 1$ for $\psi = 90°$.

### 2.3 Obscuration fraction

In geometrical solar eclipse predictions, the eclipse obscuration $f_{\text{o}}$ is commonly computed as the fraction of the solar disk occulted by the lunar disk (see e.g. Seidelmann, 1992).[3] Indeed, during an eclipse, the phase angle of the Moon approaches 180 degrees and, due to its solid composition, its near-spherical shape and its optically insignificant exosphere, the apparent eclipsing Moon can be approximated by an opaque circular disk. Not every part of the solar disk, however, contributes equally to the total solar flux, as a result of darkening of the apparent solar disk toward the solar limb, which is caused by the temperature

decrease with height in the Sun's photosphere (Chitta et al., 2020). As the Moon covers different parts of the solar disk during an eclipse, the relative contributions of the solar limb and the solar disk center to the total brightness varies. Furthermore, because the emitted radiance from the hot center peaks at shorter wavelengths than the emitted radiance from the cooler limb, the reduction of the solar irradiance during an eclipse is wavelength-dependent (Koepke et al., 2001; Bernhard and Petkov, 2019).

We use the definition of the solar limb darkening function of Koepke et al. (2001):

$$\Gamma(\lambda, r) = \frac{I_0(\lambda, r)}{I_0(\lambda, r = 0)}, \tag{4}$$

where $I_0(\lambda, r = 0)$ is the radiance originating from the solar disk center and $I_0(\lambda, r)$ is the radiance originating from the circle with radius $r$ from the solar disk center, with $r$ ranging from 0 (center) to 1 (limb). Koepke et al. (2001) parameterized the function $\Gamma$ by using the simple wavelength-dependent formula of Waldmeier (1941) based on the temperature of the Sun's

surface. We, instead, follow Ockenfuß et al. (2020) employing the parametrization of Pierce and Slaughter (1977) based on observations by the McMath-Pierce Solar Telescope, for which the limb darkening predictions showed a better agreement with

---

[3]The eclipse obscuration fraction should not be confused with the eclipse magnitude, which is the fraction of the diameter of the solar disk occulted by the Moon.

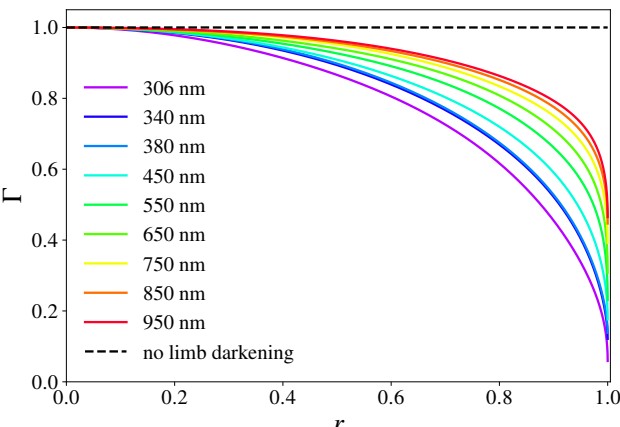

**Figure 5.** Limb darkening function $\Gamma$ for wavelengths ranging from 306 nm to 950 nm, using the limb darkening coefficients of Pierce and Slaughter (1977) and Pierce et al. (1977), as a function of distance $r$ from the solar disk center (where $r = 0$), with $r = 1$ the solar limb. The dashed line is the result without solar limb darkening taken into account ($\Gamma = 1$).

measurements of the solar spectral irradiance during the total solar eclipse on 21 August 2017 (Bernhard and Petkov, 2019). Function $\Gamma$ is computed using the 5$^{\text{th}}$ order polynomial

$$\Gamma(\lambda, r) = \sum_{k=0}^{5} a_k(\lambda) \cos^k(\psi(r)), \tag{5}$$

where $a_k$ are the limb darkening coefficients tabulated by Pierce and Slaughter (1977) for wavelengths between 303.3 nm and 729.7 nm, and by Pierce et al. (1977) for wavelengths between 740.4 and 2401.8 nm. Angle $\psi$ is the heliocentric angle as illustrated in Fig. 4 and can be computed, for any $r$ with $0 < r < 1$, from the radius of the Sun, $R_s = 695700$ km, and from the Earth-Sun distance, $d_{\text{SE}}$, which we retrieve from a geocentric ephemeris of the Sun[4]. We linearly interpolate $\Gamma$ between the tabulated wavelengths in order to compute $\Gamma$ at the wavelengths of interest. In Fig. 5, $\Gamma$ is plotted against $r$. Solar limb

darkening is most significant at the shortest wavelengths, for which the difference between the hot center and relatively cooler limb is most pronounced (see also Fig. 5 of Ockenfuß et al., 2020).

Figure 6 is a sketch of the lunar disk occulting the solar disk during an annular solar eclipse. The dimensions of the disks are normalized by the solar disk radius, such that the solar disk radius equals 1. The lunar disk radius is denoted by $r_m$. The solar disk and lunar disk centers are denoted by $C_s$ and $C_m$, respectively. Area $r d\alpha' dr$ is a differential area element of a circular ring

with radius $r$ centered at $C_s$. If no eclipse occurs, the expression for the irradiance from the solar disk, $E_0$, follows from the integration of $I_0$ (Eq. 4) over the solar disk area (Koepke et al., 2001, Eq. 2.3):

---

[4]Geocentric Ephemeris for the Sun, Moon and Planets Courtesy of Fred Espenak, http://www.astropixels.com/ephemeris/sun/sun2019.html, visited on 3 September 2020.

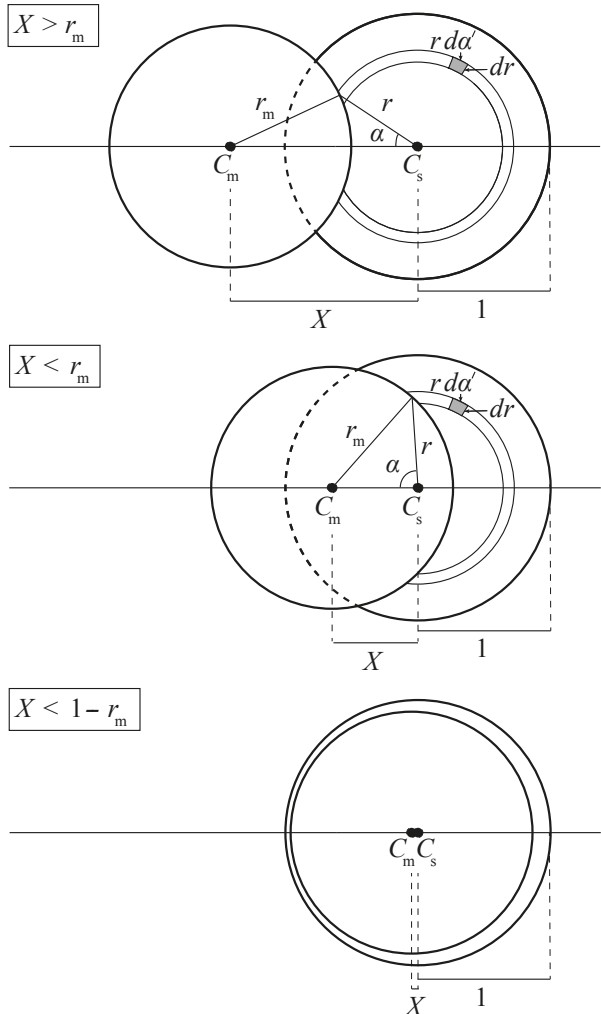

**Figure 6.** Sketches of the lunar disk (centered at $C_m$) occulting the solar disk (centered at $C_s$) during an annular solar eclipse. Here, $r_m < 1$ where $r_m$ is the radius of the lunar disk and the solar disk radius equals 1. $X$ is the distance between $C_m$ and $C_s$. For $X < r_m + 1$, the lunar disk occults the solar disk. The eclipse obscuration fraction $f_o$ increases with decreasing $X$. The annular phase occurs when $X < 1 - r_m$ (bottom sketch). Angle $\alpha$ is half the sector angle of the solar disk occulted at distance $r$ by the lunar disk.

$$
E_0(\lambda) = \int_0^1 \int_0^{2\pi} I_0(\lambda, r = 0) \cdot \Gamma(\lambda, r) \cdot r \, d\alpha' \, dr
$$

$$
= 2\pi \cdot I_0(\lambda, r = 0) \cdot \int_0^1 \Gamma(\lambda, r) \cdot r \, dr. \tag{6}
$$

During an eclipse, the irradiance from the solar disk is reduced by $f_o$, resulting from the lunar disk overlapping the solar disk. At distance $r$ from $C_s$, the angle of the sector of the solar disk that is occulted by the lunar disk is $2\alpha$ (see Fig. 6). The solar irradiance that is blocked by the Moon is

$$f_o(\lambda)E_0(\lambda) = 2 \int\limits_0^1 \int\limits_0^\alpha I_0(\lambda, r=0) \cdot \Gamma(\lambda, r) \cdot r d\alpha' dr$$

$$= 2\pi \cdot I_0(\lambda, r=0) \cdot \int\limits_0^1 \frac{\alpha(r, X, r_m)}{\pi} \Gamma(\lambda, r) \cdot r dr. \tag{7}$$

The expression for $\alpha$ follows from the geometrical consideration of the solar and lunar disks, based on $X$, $r$ and $r_m$:

$$\alpha(r, X, r_m) = \tag{8}$$

$$\begin{cases} 0 & \text{if} \quad r \leq |X - r_m| \text{ and } X > r_m, \\ \pi & \text{if} \quad r \leq |X - r_m| \text{ and } X \leq r_m, \\ \cos^{-1}\left[\frac{r^2 + X^2 - r_m^2}{2 \cdot r \cdot X}\right] & \text{if} \quad r > |X - r_m| \text{ and } r \leq X + r_m, \\ 0 & \text{if} \quad r > |X - r_m| \text{ and } r > X + r_m. \end{cases}$$

Our expression for $\alpha$ slightly differs from the one of Koepke et al. (2001), who studied a total solar eclipse ($r_m \geq 1$). Their expression is not valid during the annular phase ($X < 1 - r_m$) of an annular eclipse, where $r$ can be larger than $X + r_m$ while $r > |X - r_m|$ (cf. bottom sketch in Fig. 6), and therefore cannot be used to compute obscuration variations in the antumbra. Obscuration variations in the antumbra are most significant for annular eclipses with a relatively small $r_m$, for which the duration of the annular phase is relatively long. Equation 8 is valid in the umbra, penumbra and in the antumbra, and thus can be used during all phases of any solar eclipse type.

The eclipse obscuration fraction is computed by combining Eq. 6 and 7:

$$f_o(X, r_m, \lambda) = \frac{\int_0^1 \frac{\alpha(r, X, r_m)}{\pi} \Gamma(\lambda, r) \cdot r dr}{\int_0^1 \Gamma(\lambda, r) \cdot r dr}. \tag{9}$$

Figure 7 shows $f_o$ as a function of $X$, for wavelengths ranging from 306 nm to 950 nm, compared to the computations without limb darkening taken into account ($\Gamma = 1$), for an assumed $r_m$ of 0.97 corresponding to the instant of greatest eclipse[5] during the annular solar eclipse on 26 December 2019. The first point of contact occurs at $X = 1 + r_m = 1.97$. As the disk centers move closer to each other, $X$ decreases and $f_o$ increases. The differences with the results for $\Gamma = 1$ are again most pronounced at the shortest wavelengths (cf. Fig. 5). During the starting phase of the eclipse, the Moon occults the limb of the Sun, and not taking into account solar limb darkening results in a maximum overestimation of $f_o$ of 0.025 at 306 nm and $X = 1.52$.

---

[5]The instant of greatest eclipse is the point in time when the shadow axis passes closest to Earth's center.

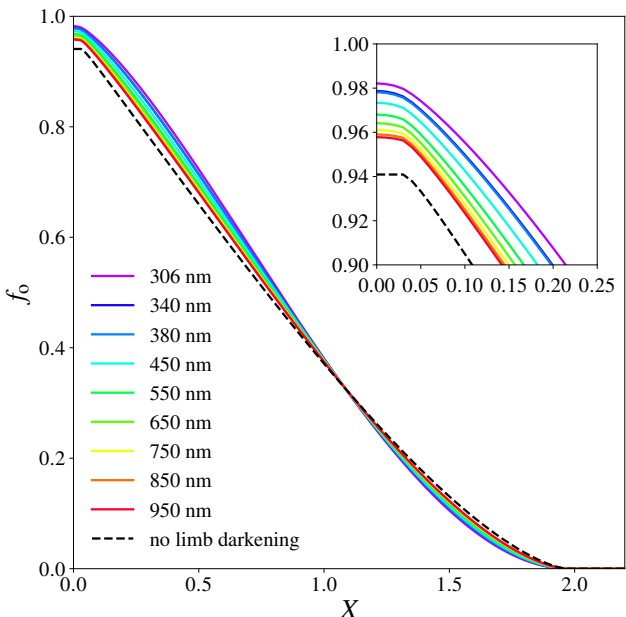

**Figure 7.** The obscuration fraction $f_o$ as a function of Moon-Sun disk center distance normalized to the solar disk radius, $X$, for wavelengths ranging from 306 nm to 950 nm, using the limb darkening coefficients of Pierce and Slaughter (1977) and Pierce et al. (1977). The dashed line is the result without solar limb darkening taken into account ($\Gamma = 1$). The assumed value for $r_m$ is 0.97, corresponding to the instant of greatest eclipse during the annular solar eclipse on 26 December 2019.

When the eclipse approaches the annular phase, the Moon occults the center of the Sun, and not taking into account solar limb darkening results in a maximum underestimation of $f_o$ of 0.069 at 306 nm and $X = 0.33$. The annular phase occurs when $X < 1 - r_m = 1 - 0.97 = 0.03$. Note that, for total eclipses, totality would occur when $X < r_m - 1$. The maximum obscuration for the ground-based observer, for a certain value of $r_m$, is reached when the centers of the lunar disk and the solar disk coincide ($X = 0$). From Eq. 8, we derive that if $X = 0$, $\alpha = \pi$ for $0 < r \leq r_m$ and $\alpha = 0$ for $r > r_m$. Given $r_m$, the maximum obscuration during an annular ($r_m < 1$) or a total ($r_m \geq 1$) eclipse is expressed by

$$f_o(X = 0, r_m, \lambda) = \begin{cases} \frac{\int_0^{r_m} \Gamma(\lambda, r) \cdot r \, dr}{\int_0^1 \Gamma(\lambda, r) \cdot r \, dr} & \text{if} \quad r_m < 1, \\ 1 & \text{if} \quad r_m \geq 1. \end{cases} \tag{10}$$

If $\Gamma = 1$, the maximum obscuration equals the area of the lunar disk divided by the area of the solar disk: $f_o(X = 0) = \pi r_m^2 / \pi = 0.941$ at the instant of greatest eclipse on 26 December 2019. At 306 nm, $f_o$ at $X = 0$ equals 0.982. Note that $f_o$ for $\Gamma = 1$ is constant within the annular phase, while the limb darkened curves (colored lines in Fig. 7) show variations of $f_o$ within the annular phase.

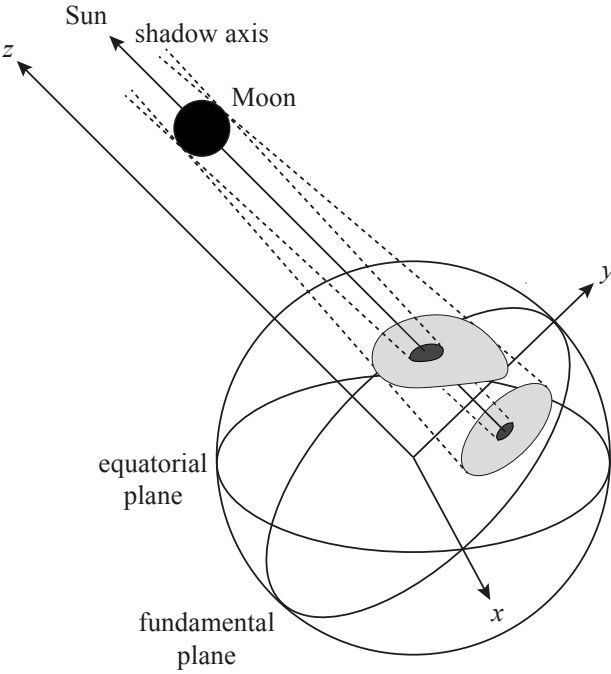

**Figure 8.** Sketch of the fundamental reference frame (not to scale). The Moon shadow that is cast on the Earth's surface has a complicated shape, but the shadow on any plane parallel to the fundamental plane ($z = 0$) has a circular shape, and the local eclipse circumstances solely depend on the distance to the shadow axis.

## 2.4 Eclipse geometry

The lunar disk radius, $r_m$, and the separation between the lunar and solar disk, $X$, depend on the location on Earth with respect to the position of the Moon and the Sun. $X$ and $r_\mathrm{m}$ can be defined for each combination of location and measurement time of a ground pixel at the Earth's surface, i.e.

$$X = X(\delta, \vartheta, h, t_1), \tag{11}$$

$$r_\mathrm{m} = r_\mathrm{m}(\delta, \vartheta, h, t_1), \tag{12}$$

where $\delta$ and $\vartheta$ are the ground pixel's geodetic latitude and longitude, respectively, $h$ is the height with respect to the Earth reference ellipsoid and $t_1$ is the measurement time belonging to the ground pixel. We transform $\delta$, $\vartheta$ and $h$ to geocentric coordinates in the so-called fundamental reference frame. The $z$-axis of the fundamental reference frame is parallel to the shadow axis, as illustrated in Fig. 8, which simplifies geometrical eclipse computations significantly. This idea was developed by Friedrich Wilhelm Bessel in the 19th century and has widely been employed to predict local circumstances of solar eclipses (Chauvenet, 1863; Meeus, 1989; Seidelmann, 1992). Even in this era of digital computers it is the most powerful eclipse

prediction technique[6]. The elements that define the orientation of the fundamental reference frame and the dimensions of the shadow are the so-called Besselian elements which are precomputed for every eclipse separately and published by NASA (Espenak and Meeus, 2006). For a certain value of $z$ in the fundamental reference frame, the local eclipse circumstances solely depend on the ground pixel's distance to the shadow-axis. In Appendix A, we provide the recipe for the computation of $X$ and $r_m$ from $\delta$, $\vartheta$, $h$ and $t_1$. We verified $r_m$ and the ground track of the shadow axis ($X = 0$) on 26 December 2019 with the eclipse predictions by Fred Espenak, NASA/Goddard Space Flight Center[7]. The mean absolute differences between our results and the NASA results for $r_m$, $\delta$ and $\vartheta$ were 0.002, 0.015° and 0.089°, respectively.[8]

## 3 Results

Here, we present the results of our computations of the eclipse obscuration fractions (Eq. 9) in the TROPOMI orbits and the corresponding restored TOA reflectance spectra (Eq. 3) during the annular solar eclipses on 26 December 2019 (Sect. 3.1) and 21 June 2020 (Sect. 3.2). With the restored TOA reflectance spectra, we correct the UV Absorbing Aerosol Index (AAI) and analyze the results. We use the example of 26 December 2019 to compare the calculated obscuration fractions to the estimated obscuration fractions from observations in an uneclipsed orbit, and to explain the AAI correction in detail. The example of 21 June 2020 is discussed more qualitatively, in which we focus on the AAI feature that we restore.

### 3.1 Annular solar eclipse on 26 December 2019

On the 26[th] of December, 2019, the Moon shadow during the annular solar eclipse followed a path along parts of Northeast Africa, Asia, and Northwest Australia. Figure 9 shows the area on the Earth's surface that was located in the penumbra (in blue) and in the antumbra (in yellow) at the instant of greatest eclipse, computed on a latitude-longitude grid with a step size of 0.05°. At the instant of greatest eclipse, the duration of the annular phase for a local observer at 1.0°N latitude and 102.2°E longitude was 3 minutes and 40 seconds, while the complete eclipse duration was 3 hours, 51 minutes and 13 seconds.[9] We compute that the penumbral shadow radius, perpendicular to the shadow axis at the Earth's surface, was 3537.3 km, while the antumbral shadow radius, perpendicular to the shadow axis at the Earth's surface, was 53.7 km. The area in the antumbra on the Earth's surface was 0.02% of the total area in the shadow of the Moon (antumbra + penumbra) on the Earth's surface.

### 3.1.1 Restored TOA reflectance

During the annular solar eclipse on 26 December 2019, TROPOMI measured the penumbra in orbit 11404 between 04:49:46 UTC and 05:48:19 UTC. The left image in Fig. 11 shows the measured TOA reflectance (without solar irradiance correction) at 380 nm on 26 December 2019, $R_{380}^{meas}$, in three adjacent orbits over Southeast Asia. An apparent decrease of $R_{380}^{meas}$ may be

---

[6]For more details, see https://eclipse.gsfc.nasa.gov/SEcat5/beselm.html, visited on 13 August 2020.

[7]See https://eclipse.gsfc.nasa.gov/SEpath/SEpath2001/SE2019Dec26Apath.html, visited on 9 October 2020.

[8]Fred Espenak rounded $r_m$ to 3 decimal digits while our results were double precision numbers. The differences in $\delta$ and $\vartheta$ were of the order of magnitude 0.01°, which was the step size of the latitude-longitude grid that we used for this verification.

[9]See https://eclipse.gsfc.nasa.gov/SEgoogle/SEgoogle2001/SE2019Dec26Agoogle.html, visited on 9 September 2020.

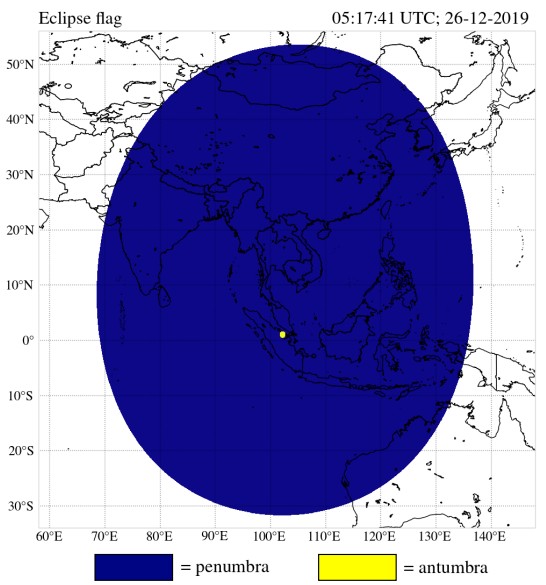

**Figure 9.** Moon shadow types at the instant of greatest eclipse during the annular solar eclipse on 26 December 2019.

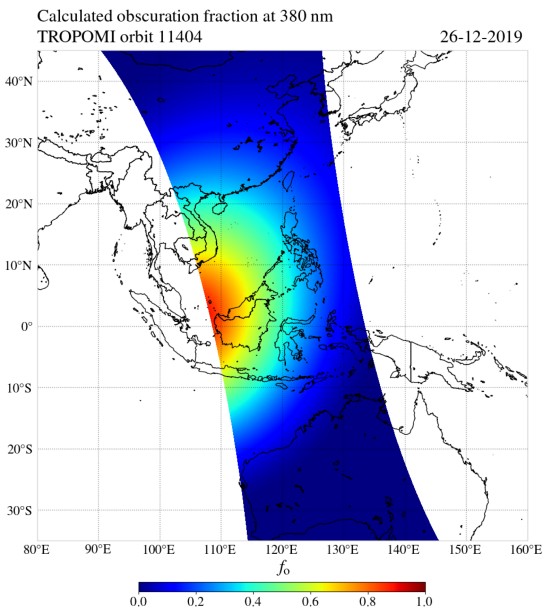

**Figure 10.** The calculated eclipse obscuration fraction at 380 nm for the ground pixels in TROPOMI orbit 11404.

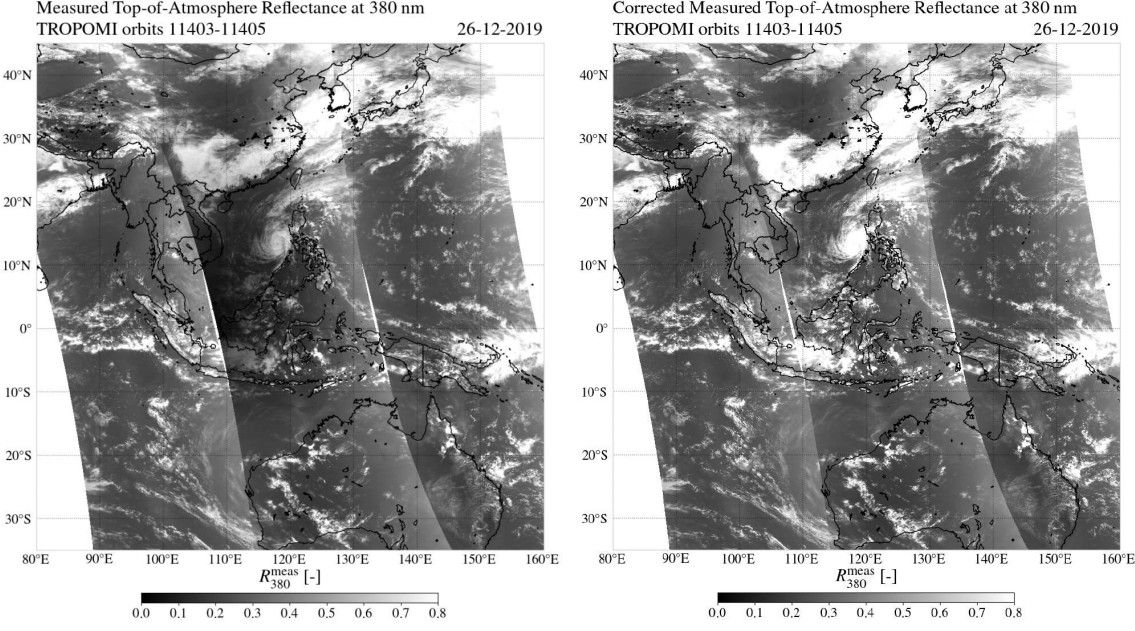

**Figure 11.** The measured top-of-atmosphere reflectance at 380 nm by TROPOMI on 26 December 2019 at Southeast Asia in orbits 11403-11405, uncorrected (left) and after the solar irradiance correction (right).

observed between 5°S and 25°N latitude, as shown by the dark shade in orbit 11404. Note that the brightening of the sky at larger viewing zenith angles, due to the increase of multiple Rayleigh scattering, is also observed in each orbit, manifesting itself in a subtle increase of $R_{380}^{\mathrm{meas}}$ toward the east and west edges of the swaths.

Figure 10 shows $f_{\mathrm{o}}$ at 380 nm that we calculated for the ground pixels of the TROPOMI UVIS detector in orbit 11404. The antumbra, in which $f_{\mathrm{o}}$ at 380 nm peaks at 0.976 (see Fig. 7), was not captured because the antumbra was located slightly out of sight in the West. The maximum calculated $f_{\mathrm{o}}$ for this orbit is 0.89 at 2.27°N latitude and 108.12°E longitude. Figure 10 shows that the eclipse obscuration in orbit 11404 was not limited to the Gulf of Thailand and the South China Sea: small obscuration fractions ($0 < f_{\mathrm{o}} < 0.4$) could be experienced in Eastern China and the Northwest coast of Australia.

The right image in Fig. 11 shows the restored TOA reflectance at 380 nm, that is, after the correction for the eclipse obscuration (Eq. 3) in orbit 11404. The dark shade that could be observed in the left image in Fig. 11, resulting from the decreased $R_{380}^{\mathrm{meas}}$ in the Moon shadow, has disappeared. The appearance of the corrected $R_{380}^{\mathrm{meas}}$ in orbit 11404 is comparable to the appearance of $R_{380}^{\mathrm{meas}}$ in orbits 11403 and 11405.

To analyze the reflectance correction more quantitatively, Fig. 12 shows the average $R_{380}^{\mathrm{meas}}$ per scanline[10] in orbit 11404 against the mean latitude in the scanline (i.e., the average of all pixel rows), and the corresponding average calculated $f_{\mathrm{o}}$ at 380 nm. The dotted line represents $R_{380}^{\mathrm{meas}}$ before the solar irradiance correction, and the solid line represents $R_{380}^{\mathrm{meas}}$ after the solar

---

[10]The line at the Earth's surface perpendicular to the flight direction defined by the satellite swath which is roughly oriented West-East.

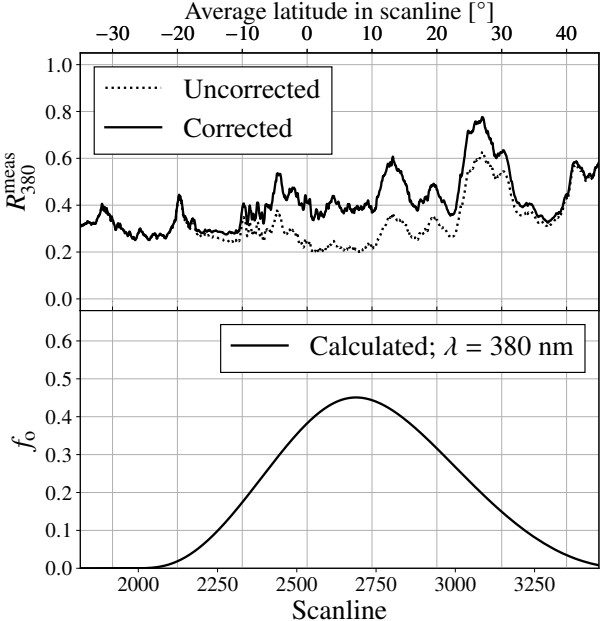

**Figure 12.** The average solar irradiance corrected (solid line) and uncorrected (dotted line) measured top-of-atmosphere reflectance at 380 nm by TROPOMI at 26 December 2019 in orbit 11404 per scanline (top image), and the corresponding average calculated obscuration fraction $f_{\mathrm{o}}$ at 380 nm (bottom image).

irradiance correction. The peak in both curves at 14°N is caused by the spiral cloud deck between Vietnam and the Philippines, and the peak at 25°N latitude is caused by the cloud deck above Southeast China (cf. Fig. 11). Before the solar irradiance correction, the lowest values are measured where $f_o$ is highest, between 3°S and 10°N latitude. After the solar irradiance correction, the $R_{380}^{meas}$ curve is increased, but only at the latitudes where the Moon shadow resided.

### 3.1.2 Comparison to the observed obscuration fraction

The restored TOA reflectance during an eclipse that we showed in Sect. 3.1.1 can be considered the intrinsic reflectance $R^{int}$ of the atmosphere-surface system, as explained in Sect. 2.1. If the optical properties of the atmosphere-surface system are not affected by the eclipse, $R_{int}$ approximates the TOA reflectance as if there were no eclipse. Consequently, the eclipse obscuration $f_o$ at 380 nm can be estimated from the comparison of observations of $R_{380}^{meas}$ inside and outside the Moon shadow, and can be used to verify the calculated $f_o$ at 380 nm from theory.

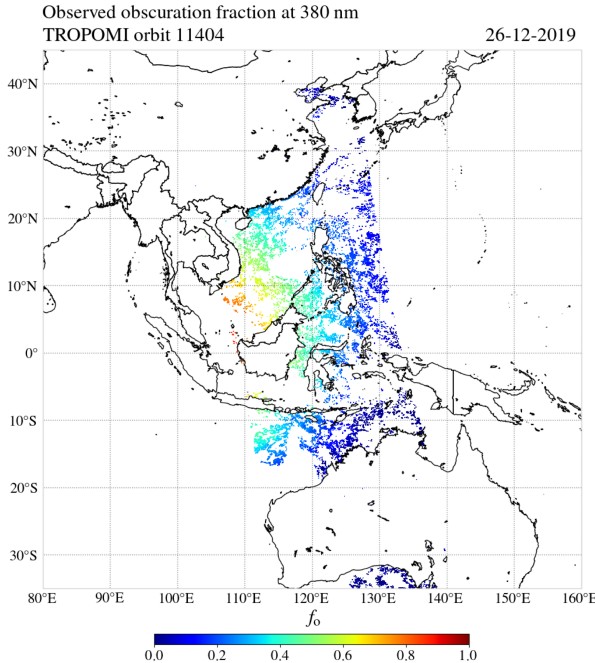

**Figure 13.** The observed eclipse obscuration fraction at 380 nm for the ground pixels in TROPOMI orbit 11404 that passed the filters described in Sect. 3.1.2.

Orbit 11403 (east of orbit 11404) was not eclipsed and preceded the eclipsed orbit 11404. We compare each ground pixel of orbit 11404 to its equivalent in orbit 11403, i.e., for the same scanline and pixel row, such that differences in illumination and viewing geometry are negligible. That is, we compute the observed $f_o$ at 380 nm as (cf. Eq. 3)

$$f_o(\lambda = 380 \text{ nm}) \approx 1 - \frac{[R_{380}^{\text{meas}}]_{\text{eclipse}}}{[R_{380}^{\text{meas}}]_{\text{no eclipse}}}, \tag{13}$$

where the label 'eclipse' indicates orbit 11404 and the label 'no eclipse' indicates orbit 11403. We can compute Eq. 13 for pixels that have a comparable atmosphere-surface system. Therefore, we only compare ocean pixels, because, at the latitudes where the eclipse was measured in orbit 11404, orbit 11403 was mainly above the Pacific Ocean. Also, we only consider cloud-free pixels as the cloud types and cloud fractions in the two pixels will hardly be identical. In Fig. 7 we showed that the difference of $f_o$ between 340 nm and 380 nm is insignificant (the $f_o$ curves for 340 nm and 380 are virtually indistinguishable), so the ratio

$R_{340}^{\text{meas}}/R_{380}^{\text{meas}}$ should not be affected by the eclipse for a constant atmosphere-surface system. That is, if the atmosphere-surface system of the pixel in orbit 11404 is approximately identical to the atmosphere-surface system of its equivalent pixel in orbit 11403, the ratio $R_{340}^{\text{meas}}/R_{380}^{\text{meas}}$ is expected to be approximately identical regardless of the eclipse conditions. Before estimating $f_o$ from observations, we therefore apply the filter

$$\left| \left[ \frac{R_{340}^{\text{meas}}}{R_{380}^{\text{meas}}} \right]_{\text{eclipse}} - \left[ \frac{R_{340}^{\text{meas}}}{R_{380}^{\text{meas}}} \right]_{\text{no eclipse}} \right| < 0.01. \tag{14}$$

Some cloudy pixels may pass the filter of Eq. 14, because clouds can alter the TOA reflectance spectra at both 340 nm and 380 nm. The cloud fraction product FRESCO (Koelemeijer et al., 2001; Wang et al., 2008) is available on the TROPOMI UVIS grid, but suffers from the eclipse. For this comparison, we apply the simple cloud filter

$$R_{340}^{\text{meas}} \cdot 0.95 > R_{380}^{\text{meas}} \tag{15}$$

to both orbits, which deletes the majority of the pixels with thick bright clouds. This filter is based on the fact that the TOA
reflectance over the cloud-free ocean generally decreases with increasing wavelength from 340 nm to 380 nm (see e.g. Tilstra et al., 2020, Fig. 1), while the presence of clouds may increase the TOA reflectance spectrum toward 380 nm.

Figure 13 shows the observed $f_o$ at 380 nm, computed with Eq. 13, that passed the filters described in this section. Note the good agreement with the calculated $f_o$ at 380 nm in Fig. 10. The missing values result mostly from land or cloudy pixels in orbit 11403 or 11404. Between 0°N and 10°N latitude, at the very west side of the swath in orbit 11404, many pixels did
not pass the filter of Eq. 14, which can be explained by the thin clouds that were present (see Fig. 11), but also possibly by a difference in aerosol type and concentration or ocean color with respect to the pixels in orbit 11403.

Figure 14 shows the calculated $f_o$ at 380 nm (solid line) and the observed $f_o$ at 380 nm (diamond dots) for pixel row 6 (out of 450, i.e., at the west side of the swath) and scanline 2000 to 3500. The dashed line is the calculated $f_o$ when solar limb darkening is not taken into account ($\Gamma = 1$). Taking into account limb darkening in the calculation of $f_o$ results in a much

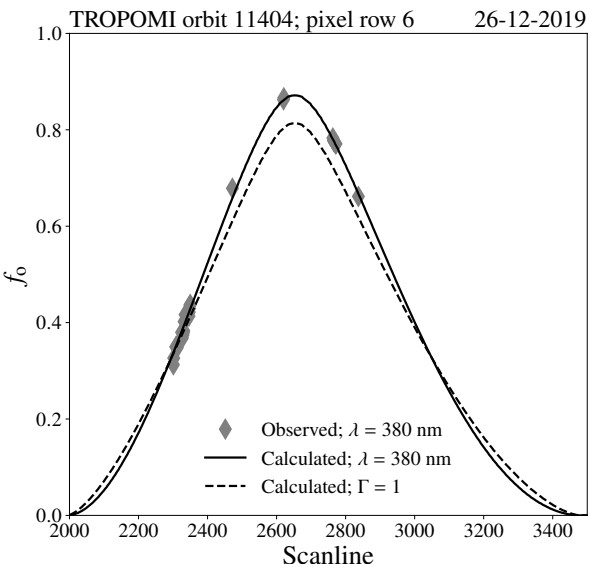

**Figure 14.** The observed eclipse obscuration fraction by TROPOMI in orbit 11404 on 26 December 2019 (grey diamond dots), compared to the calculated eclipse obscuration fraction at 380 nm including solar limb darkening (black solid line) and for $\Gamma = 1$ which excludes solar limb darkening (black dashed line), per scanline in pixel row 6.

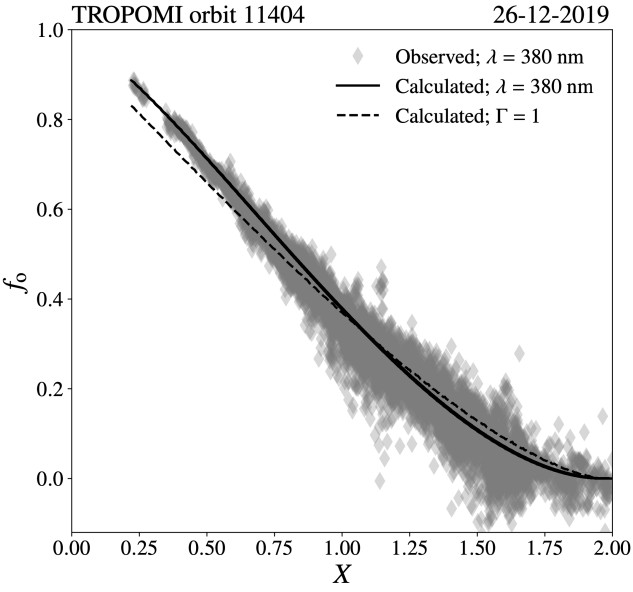

**Figure 15.** Similar to Fig. 14, but plotted against the disk center separation $X$.

better agreement with the observed $f_o$ at 380 nm. This can also be concluded from Fig. 15, where we show the calculated and observed $f_o$ at 380 nm for all pixels in Fig. 13 plotted against the Moon-Sun disk center distance normalized to the solar disk radius, $X$, computed for each of those pixels.[11] On the domain $X < 0.5$, the total mean absolute difference between the observed and calculated $f_o$ at 380 nm was 0.008, while the total mean absolute difference between the observed and the calculated $f_o$ for $\Gamma = 1$ was 0.053. The maximum underestimation of $f_o$ at 380 nm when using $\Gamma = 1$, with respect to $f_o$ at 380 nm when solar limb darkening is taken into account, was 0.06 at 6.04°N latitude and 107.19°E longitude.

### 3.1.3 Restored UV Absorbing Aerosol Index

The AAI as derived by TROPOMI is computed from the ratio of the measured reflectances at 340 and 380 nm and the ratio of the modeled reflectances at those wavelengths, according to (Herman et al., 1997; Torres et al., 1998)

$$\text{AAI} = -100 \cdot \left[ \log_{10} \left( \frac{R_{340}}{R_{380}} \right)^{\text{meas}} - \log_{10} \left( \frac{R_{340}}{R_{380}} \right)^{\text{model}} \right], \tag{16}$$

where 'meas' indicates the measured TOA reflectances and 'model' indicates the modeled TOA reflectances. The modeled TOA reflectances are computed for a cloud-free and aerosol-free atmosphere-surface model with the 'Doubling-Adding KNMI' (DAK) radiative transfer code (de Haan et al., 1987; Stammes, 2001), version 3.1.1, taking into account single and multiple Rayleigh scattering and absorption of sunlight by molecules in a pseudo-spherical atmosphere, including polarization. The Lambertian surface albedo $A_s$ in the model is assumed independent of wavelength $\lambda$ and is adjusted such that the model reflectance equals the measured reflectance at 380 nm:

$$R_{380}^{\text{model}}(A_s) = R_{380}^{\text{meas}}. \tag{17}$$

The value of $A_s$ that satisfies Eq. 17 is often referred to as the 'scene albedo' or the 'Lambertian equivalent reflectance (LER)'. Because $A_s$ is assumed wavelength independent, it is also used to compute $R_{340}^{\text{model}}$. More details about the TROPOMI AAI algorithm can be found in Stein Zweers et al. (2018). For our solar eclipse application, it should be noted that a lower $R_{380}^{\text{meas}}$ results in a smaller (spectrally flat) surface contribution in the model, which increases $R_{340}^{\text{model}}/R_{380}^{\text{model}}$ and increases the AAI.

The UV Absorbing Aerosol Index (AAI) can be interpreted as a comparison of the measured TOA reflectance UV color to the TOA reflectance UV color of a cloud-free and aerosol-free atmosphere-surface model. The AAI generally increases in the presence of absorbing aerosols and can, unlike the aerosol optical depth, also be computed when the aerosol layer is above clouds. In the next paragraph, we provide a brief introduction to the AAI. The AAI depends on various parameters such as the aerosol optical depth (AOD), single scattering albedo (SSA) and aerosol layer height (ALH). For more details about the

---

[11]The density of points increases with increasing $X$ because the Earth's surface area for which a certain value of $X$ applies increases with increasing $X$. Making the filter of Eq. 15 more strict (e.g. $R_{340}^{\text{meas}} > 0.75 \cdot R_{380}^{\text{meas}}$), decreases the scatter but also decreases the number of points. Another reason for the increasing scatter with increasing $X$ is that for low $f_o$ the compared pixels may have more differences, resulting from natural variations, than caused the by the obscuration ($\left[ R_{380}^{\text{meas}} \right]_{\text{eclipse}} / \left[ R_{380}^{\text{meas}} \right]_{\text{no eclipse}}$ in Eq. 13 and the impact of its variations on $f_o$ are relatively large).

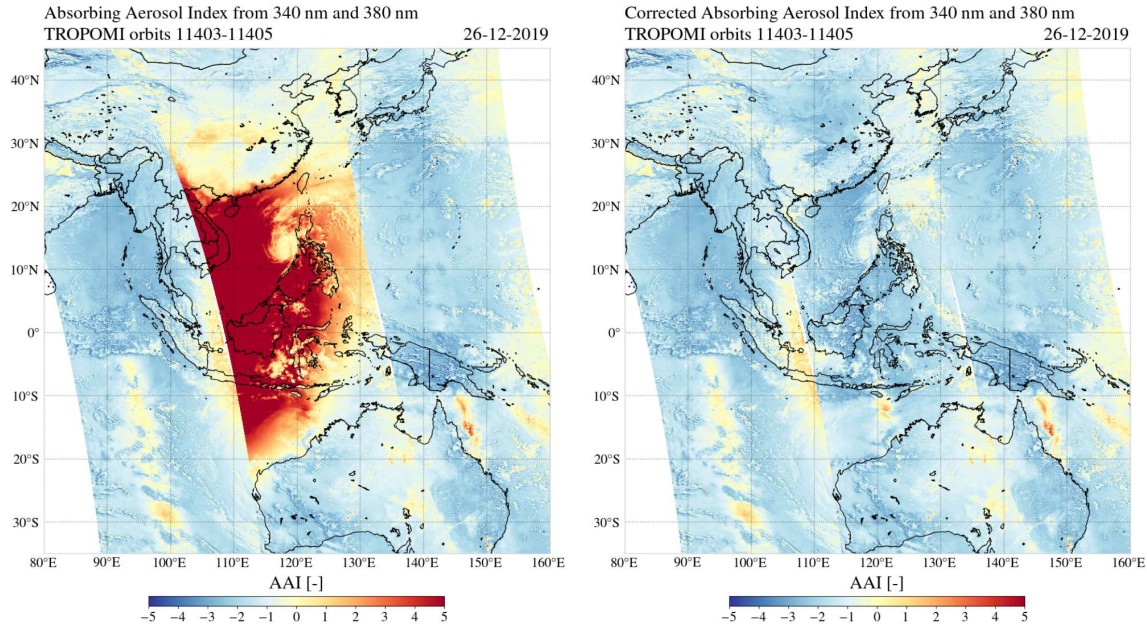

**Figure 16.** The Absorbing Aerosol Index from the 340/380 nm wavelength pair by TROPOMI on 26 December 2019 at Southeast Asia in orbits 11403-11405, uncorrected (left) and after the solar irradiance correction (right).

sensitivity of the AAI to atmosphere and surface parameters, we refer to Herman et al. (1997), Torres et al. (1998), de Graaf et al. (2005), Penning de Vries et al. (2009) and Kooreman et al. (2020). In Appendix B we provide an analysis of the precision of the AAI during the solar eclipses studied in this paper.

The left image in Fig. 16 shows the AAI measured by TROPOMI during the annular solar eclipse on 26 December 2019,
in the three adjacent orbits over Southeast Asia considered in Sect. 3.1.1. We use a color scale ranging from $AAI = -5$ to $AAI = 5$. This range usually covers most aerosol events. Significantly elevated AAI values are measured at the location of the penumbra: in orbit 11404, most apparent between $20°S$ and $30°N$ latitude (cf. $f_o$ in Fig. 10). The maximum AAI was 53.9 at $2.17°N$ latitude and $108.14°E$ longitude. The mean AAI in orbit 11404 was 0.15. At the spiral cloud deck centered at $15°N$ latitude and $118°E$ longitude, the AAI increase is less significant. Similarly, the clouds observed in Fig. 11 between $10°S$ and
$0°N$ latitude, and between $22°N$ and $32°N$, are located in the penumbra but show a less significant AAI increase.

Outside the Moon shadow, in orbits 11403 and 11405, the mean AAI is $-1.52$ and $-1.48$, respectively. At the locations in orbit 11404 where $f_o < 0.2$, the mean AAI is also negative ($\sim -1.5$). The negative mean AAI are partly caused by the scattering of light by cloud droplets, but also due to a radiometric calibration offset and degradation in the TROPOMI irradiance data (Tilstra et al., 2020; Ludewig et al., 2020). The degradation in the irradiance leads to an increase of the derived reflectance,
decreasing the AAI values over time. The total AAI bias of $\sim -1.5$ will be solved with the release of the version 2.0.0 TROPOMI level 1b processor, foreseen for the first half of 2021. The bias is expected to be independent of viewing geometry, hence, it will not affect the relative AAI values nor the conclusions of this paper.

The right image in Fig. 16 is similar to the left image in Fig. 16, but then for the corrected AAI product. That is, in Eq. 16 and 17, we replaced the measured TOA reflectances, $R_{340}^{\mathrm{meas}}$ and $R_{380}^{\mathrm{meas}}$, by the restored TOA reflectances, $R_{340}^{\mathrm{int}}$ and $R_{380}^{\mathrm{int}}$, which we computed with Eq. 3. The red spot between $20°$S and $30°$N in orbit 11404 that was observed in the uncorrected AAI product has disappeared. The mean of the corrected AAI in orbit 11404 is $-1.58$. At the location of the thick spiral cloud deck the AAI is closer to zero. We note that no significant absorbing aerosol events can be identified in Figure 16. At $12°$S latitude and $122°$E longitude, an AAI increase is measured in the corrected product, which could not be observed in the uncorrected image. This feature is caused by the specular reflection off the sea surface, often called the sunglint (see also Fig. 11). The sunglint can also be observed in the middle of the swath of orbits 11403 and 11405, between $20°$S and $10°$S latitude and $34°$S and $5°$S latitude, respectively. Kooreman et al. (2020) explain that, when a strongly anisotropic reflector such as the sea surface is viewed from its reflective side, the AAI may increase: the model assumes a Lambertian (isotropic reflecting) surface, which increases the relative importance of the Rayleigh scattered light in the model and therefore computes a higher $R_{340}/R_{380}$ than is measured. From Eq. 16, it follows that a deficit in the measured $R_{340}/R_{380}$ results in an increased AAI. Note that the shape and size of the apparent sunglint may vary per orbit, as they depend on the roughness of the sea surface (i.e. the wind speed), the presence of clouds and aerosols, and the illumination and viewing geometries.

In Fig. 17, we show the average $R_{340}^{\mathrm{meas}}/R_{380}^{\mathrm{meas}}$, $R_{340}^{\mathrm{model}}/R_{380}^{\mathrm{model}}$, $A_{\mathrm{s}}$ and AAI of the pixels in the scanlines of orbit 11404, before the solar irradiance correction (dotted line) and after the solar irradiance correction (solid line). The average latitudes in the scanlines are also shown. The fraction $R_{340}^{\mathrm{meas}}/R_{380}^{\mathrm{meas}}$ is not affected by the solar irradiance correction, which is expected from the wavelength independence of $f_{\mathrm{o}}$ between 340 and 380 nm (see Fig. 7). Here, we did not detect signatures of sky color changes in the measured UV reflectance due to secondary effects such as horizontally travelled light (see Sect. 4, for a detailed discussion). Before the solar irradiance correction, $R_{340}^{\mathrm{model}}/R_{380}^{\mathrm{model}}$ is significantly higher than $R_{340}^{\mathrm{meas}}/R_{380}^{\mathrm{meas}}$ between $20°$S and $30°$N where the Moon shadow resided, which increases the AAI. The high $R_{340}^{\mathrm{model}}/R_{380}^{\mathrm{model}}$ is caused by the relatively low $A_s$ in the Moon shadow (Fig. 17), which is caused by the decrease of $R_{380}^{\mathrm{meas}}$ by $f_{\mathrm{o}}$ (Fig. 12). The maximum scanline average AAI is 10.7 in scanline 2672 (see bottom graph in Fig. 17). After the solar irradiance correction, the AAI increase in the Moon shadow disappeared because $R_{340}^{\mathrm{model}}/R_{380}^{\mathrm{model}}$ follows an approximately similar pattern as $R_{340}^{\mathrm{meas}}/R_{380}^{\mathrm{meas}}$, albeit with an offset ranging from $-0.03$ to $-0.06$, which was also observed outside the Moon shadow. We conclude that the increased AAI between $20°$S and $30°$N latitude in orbit 11404 before the solar irradiance correction was caused by the relatively low $A_{\mathrm{s}}$ used in the model due to the reduction of the measured reflectance at 380 nm, rather than a UV color change of the measured TOA reflectance in the Moon shadow.

Figure 18 shows the AAI in the scanlines of orbit 11404, but only for pixel row 6 (cf. Fig. 14). The solid line is the eclipse corrected AAI and the dotted line is the eclipse corrected AAI but without taken into account limb darkening ($\Gamma = 1$). If solar limb darkening is not taken into account, the corrected AAI still shows an apparent increase between $14.9°$S and $21.6°$N latitude, with a maximum of AAI $= 4.3$ at $3.37°$N latitude. Note that these are the latitudes at which the $f_{\mathrm{o}}$ was underestimated if $\Gamma = 1$ as we showed in Fig. 14, caused by the Moon occulting different parts of the solar disk during the eclipse (see Sect. 2.3). In line with the discussion of the previous paragraph, an underestimation of $f_{\mathrm{o}}$ at 380 nm results, after the solar irradiance correction, in too low $R_{380}^{\mathrm{meas}}$ and $A_{\mathrm{s}}$, in too high $R_{340}^{\mathrm{model}}/R_{380}^{\mathrm{model}}$ and, therefore, in too high AAI. We find a maximum

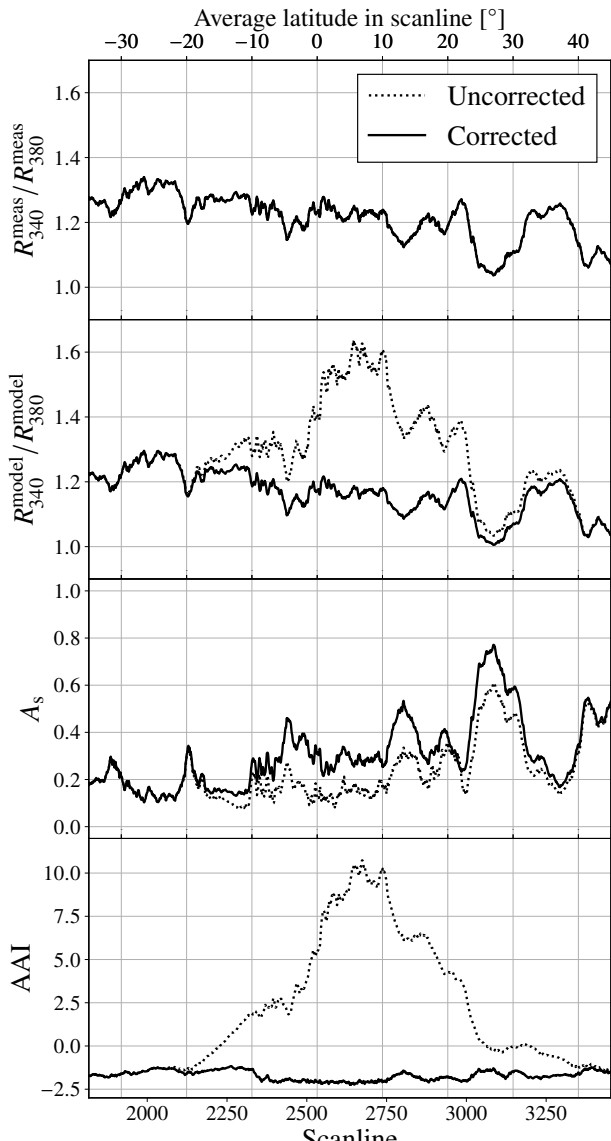

**Figure 17.** The scanline averages of (from top to bottom) $R_{340}^{\mathrm{meas}}/R_{380}^{\mathrm{meas}}$, $R_{340}^{\mathrm{model}}/R_{380}^{\mathrm{model}}$, $A_\mathrm{s}$ and AAI in orbit 11404, 26 December 2019. The average latitudes in the scanlines are indicated at the top. The dotted lines are the results before the solar irradiance correction and the solid lines are the results after the solar irradiance correction. The lines for the uncorrected and corrected $R_{340}^{\mathrm{meas}}/R_{380}^{\mathrm{meas}}$ overlap.

overestimation of the AAI of 6.7 points in scanline 2671 and pixel row 4 when using $\Gamma = 1$. It can be concluded that not taking into account solar limb darkening would still result in a 'red spot' anomaly in the AAI map. The opposite effect occurs at the

375 latitudes where $f_\mathrm{o}$ was overestimated if $\Gamma = 1$ in Fig. 14: south from $14.9°$S and north from $21.6°$N latitude, the AAI after the correction without limb darkening is slightly lower than the AAI after the correction where limb darkening was taken into

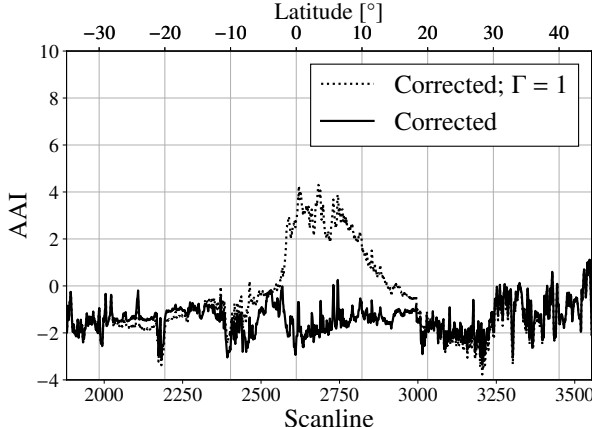

**Figure 18.** The AAI in pixel row 6 in orbit 11404, 26 December 2019. The latitudes in the scanlines are indicated at the top. The solid line is the result after the solar irradiance correction, the dotted line is the result after the solar irradiance correction when solid limb darkening is not taken into account ($\Gamma = 1$).

account. We conclude that, if the artificial Moon shadow signatures are to be removed in the corrected AAI product, solar limb darkening cannot be neglected.

### 3.2 Annular solar eclipse on 21 June 2020

On the 21$^{\text{st}}$ of June, 2020, the Moon shadow during an annular solar eclipse could be experienced in the majority of Africa (from South Africa to Libya) and almost all parts of Asia. At the instant of greatest eclipse, $r_{\text{m}}$ was 0.994 and the duration of the annular phase for a local observer at 30.5°N latitude and 80.0°E longitude was 38 seconds, while the complete eclipse duration was 3 hours, 26 minutes and 53 seconds.[12] We compute that the penumbral shadow radius, perpendicular to the shadow axis at the Earth's surface, was 3493.9 km, while the antumbral shadow radius, perpendicular to the shadow axis at the Earth's

surface, was 10.5 km. The area in the antumbra on the Earth's surface was 0.0008% of the total area in the shadow of the Moon (antumbra + penumbra) on the Earth's surface.

Figure 19 shows $R_{380}^{\text{meas}}$ by TROPOMI on 21 June 2020 over Asia, before the solar irradiance correction (left image) and after the solar irradiance correction (right image). The shadow of the Moon was captured in orbit 13930, as shown by the apparent decrease of $R_{380}^{\text{meas}}$ between 10°N and 55°N latitude. Only the penumbra was captured. The antumbra was located out of sight

in the west of orbit 13930. The maximum calculated $f_{\text{o}}$ at 380 nm was 0.92 at 31.94°N latitude and 82.51°E longitude.

The left image in Fig. 20 shows the TROPOMI AAI in orbits 13929 to 13931 over Asia, before the solar irradiance correction. Before the solar irradiance correction, the AAI is significantly increased in the shadow of the Moon. The edge of the red spot in the uncorrected AAI in orbit 13930 is shaped by the local cloudiness. For example, the AAI > 2 signature in West Mongolia (40-50°N latitude and 90-110°E longitude), is spatially correlated with low cloud fraction area. Note that this suppression of

---

[12]See https://eclipse.gsfc.nasa.gov/SEgoogle/SEgoogle2001/SE2020Jun21Agoogle.html, visited on 28 September 2020.

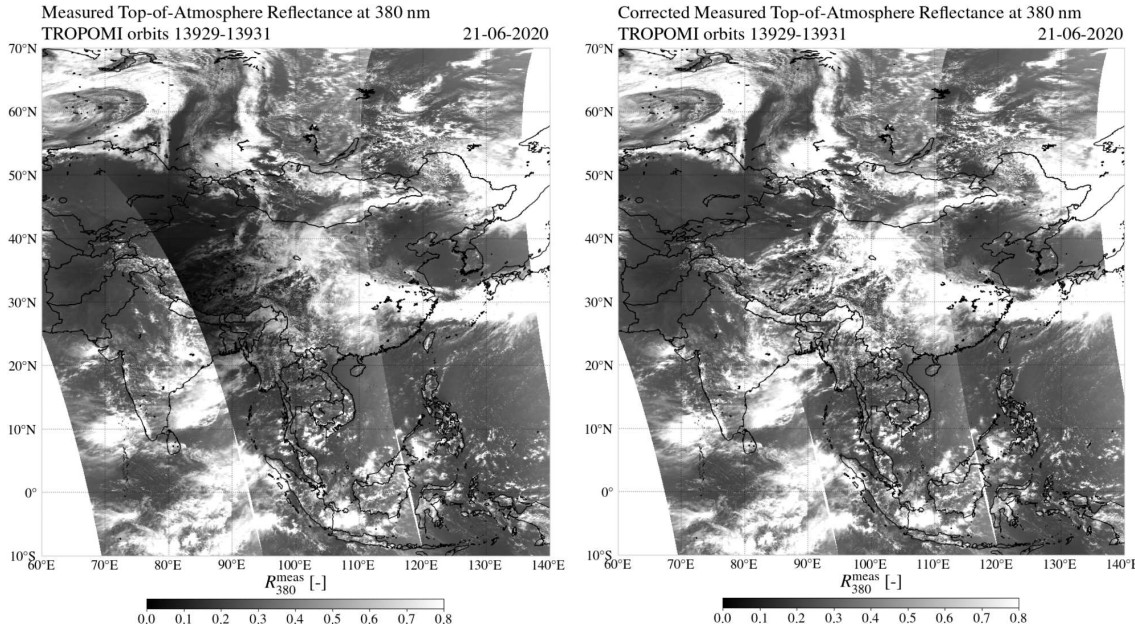

**Figure 19.** The measured top-of-atmosphere reflectance at 380 nm by TROPOMI on 21 June 2020 over Asia in orbits 11403-11405, uncorrected (left) and after the solar irradiance correction (right).

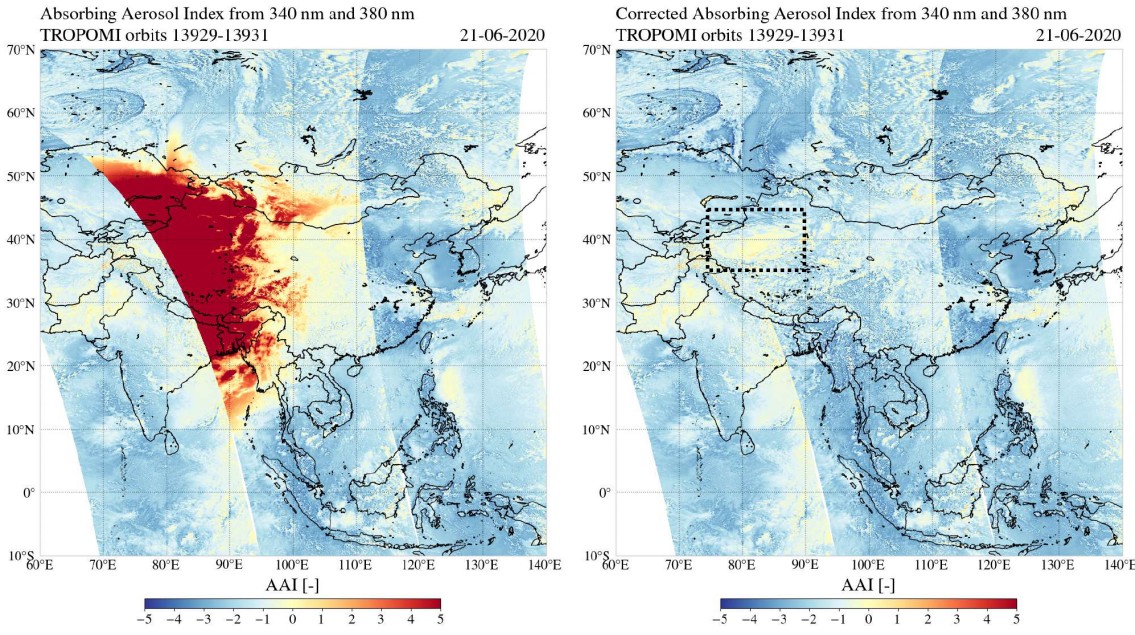

**Figure 20.** The Absorbing Aerosol Index from the 340/380 nm wavelength pair by TROPOMI on 21 June 2020 over Asia in orbits 13929-13931, uncorrected (left) and after the solar irradiance correction (right). In the corrected image, the Taklamakan desert is located in the rectangular dotted box.

the eclipse anomaly by clouds in the AAI product was also observed at the cloudy areas in the shadow of the Moon on 21 December 2019 (left image of Fig. 16).

The right image in Fig. 20 is similar to the left image in Fig. 20, but after the solar irradiance correction. The red spot between 10°N and 55°N latitude in orbit 13930 that was observed in the uncorrected AAI product has disappeared. In Northwest China, a region of relatively high AAI values appears in the corrected product: at 36°-42°N latitude and 78°-86°E longitude, the AAI is increased by ∼ 1.5 points. Note that this AAI change is larger than the maximum standard AAI error in orbit 13930 of 0.40 (see Appendix B). We verify this AAI feature using AAI measurements of the Global Ozone Monitoring Experiment–2 (GOME-2) instrument on board the Metop-C satellite (referred to as 'GOME-2C' in what follows). Figure 21 shows the AAI measured by GOME-2C on 21 June 2020 in orbits 8411 to 8415 in the Middle-East and Western Asia, from the Polarization Measurement Detectors (PMDs) using $\lambda = 338$ nm and $\lambda = 381$ nm for the AAI retrieval (see Tuinder et al., 2019). In order to show the eclipse location during the GOME-2C measurements, we did not apply the solar irradiance correction to the GOME-2C data. Figure 21 shows that two GOME-2C orbits were affected by the eclipse: significantly elevated AAI were measured in the shadow of the Moon in orbit 8413 and 8414. The location 36°-42°N latitude and 78°-86°E longitude was not eclipsed during the measurements of GOME-2C. Indeed, GOME-2C also measured an AAI increase of ∼ 1.5 points in this same area in Northwest China. At this location, the Taklamakan Desert is located. The Taklamakan desert is the largest desert in China, about 960 km long and 420 km wide, and consists mostly of shifting sand dunes that reach elevations of 800 m to 1500 m

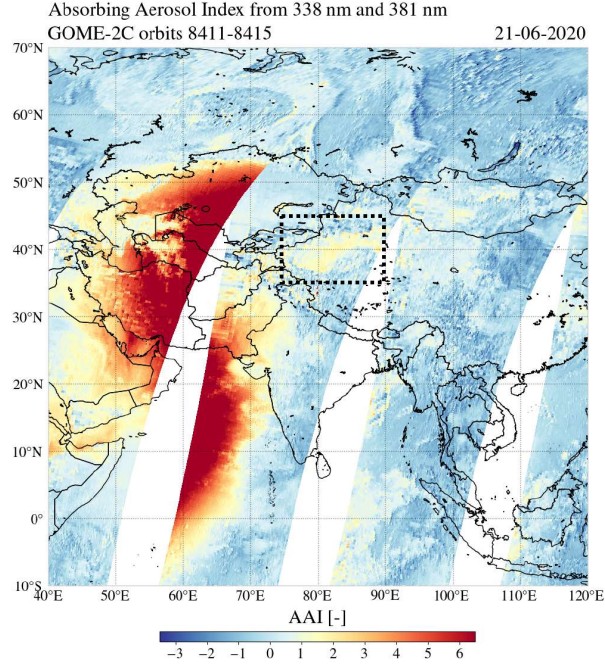

**Figure 21.** The Absorbing Aerosol Index from the 338/381 nm wavelength pair by GOME-2C on 21 June 2020 over Asia in orbits 8411-8415. The Taklamakan desert is located in the rectangular dotted box.

above sea level (Petrov and Alitto, 2019). It is an important source for the global atmospheric dust budget and for dust storms in Eastern Asia (Hu et al., 2020). Hence, we attribute this $\sim 1.5$ points increase to the desert surface and, possibly, desert dust aerosol.

## 4  Discussion

The eclipse obscuration theory provided in Sect. 2.3 applies to any phase of any solar eclipse type. The TROPOMI orbits during the annular solar eclipses analyzed in this paper did not capture the antumbra. The maximum $f_{\mathrm{o}}$ at 380 nm calculated in the TROPOMI orbits were 0.89 and 0.92 on 26 December 2019 and 21 June 2020, respectively, while the annular phase for these eclipses at 380 nm occurred for $f_{\mathrm{o}} > 0.976$ and $f_{\mathrm{o}} > 0.997$, respectively. In this section, we reflect back on the assumptions we made and we discuss some points of attention for potential future applications of the solar irradiance correction
to measurements closer to the (ant)umbra, and/or in the antumbra.

In this paper, we assumed that the solar irradiance is randomly polarized. Sunlight scattered in the Sun's atmosphere may become polarized. This linearly polarized spectrum is also known as the 'Second Solar Spectrum' and its significance increases towards the solar limb (Stenflo and Keller, 1997). If there is no eclipse, this polarization cancels out due to the symmetry and only very small linear degrees of polarization of the disk-integrated sunlight can be measured (on the order of $10^{-6}$, see
Kemp et al. (1987), who attributed this polarization in their ground-based observations to multiple scattering in the Earth's atmosphere). During an eclipse, the symmetry is broken, however, measurements show that a few arcseconds inside the solar limb the degree of polarization is lower than 0.01 and in most cases less than 0.001 (Stenflo, 2005)[13]. Only at $0.99 < r < 1$, the degree of polarization can be larger than 0.01, but never will grow bigger than the theoretical limit of 0.117 (Stenflo, 2005). Note that, during the annular solar eclipse of 26 December 2019, $r_{\mathrm{m}} = 0.97$ and the minimum disk center separation
$X$ in the TROPOMI pixels was 0.22, meaning that, at the straight line through the solar disk and lunar disk centers, the most narrow visible solar limb (when the opposite solar limb was occulted) was $0.75 < r < 1$. For 21 June 2020, $r_{\mathrm{m}} = 0.994$ and the minimum encountered $X$ was 0.20, giving $0.79 < r < 1$. Hence, it may be expected that, integrated over the visible solar disk, the polarization of the sunlight was negligible.

For this paper we did not take into account light travelling horizontally in the atmosphere from one ground pixel to the other.
A well-known phenomenon during total solar eclipses, for a local observer in the umbra, is the reddening of the sky near the horizon, i.e., of light scattered from outside the umbra (Shaw, 1975). The path lengths of scattered beams reaching a ground sensor in the umbra are relatively long, i.e., Rayleigh scattering may cause the reddening near the horizon, while overhead the sky may appear more blue (Gedzelman, 1975). 3-D radiative transfer code simulations by Emde and Mayer (2007) suggest that scattered horizontal visible irradiance reaching a ground sensor in the umbra is about 20000 times smaller at 330 nm and about
23000 smaller at 500 nm than the total (direct + diffuse) irradiance received in uneclipsed conditions. Outside the umbra where, for example, $f_{\mathrm{o}} < 0.99$, already $> 1\%$ of the uneclipsed solar irradiance is received at TOA which is expected to dominate the

---

[13]See also https://ethz.ch/content/dam/ethz/special-interest/phys/particle-physics/cosmologygroup-dam/People/StenfloPDFs/stenflo_spse06.pdf, visited on 8 October 2020.

horizontally travelled light. Emde and Mayer (2007) compared their 3-D simulations to a 1-D approach and computed that longer than 10 minutes before or after totality the uncertainty of a 1-D method is lower than 1%. An analysis of this 1-D bias for the TOA reflectance versus $f_o$ could give a definite limit in terms of $f_o$. In the TROPOMI orbits studied in this paper, the maximum calculated $f_o$ at 380 nm were 0.89 and 0.92, which explains why we did not detect anomalies in the restored TOA reflectances, nor in the corrected AAI, that could be attributed to 3-D effects or a reddening of the measured UV spectrum.

A second reason for the potential reddening of the sky during a solar eclipse has a fundamentally different origin. Yellow and orange cloud tops have been observed, for example, in true color MODIS satellite images in the penumbra during the total eclipse of 2 July 2019 and during the annular solar eclipse on 26 December 2019 (Gedzelman, 2020). During the total solar eclipse of 20 March 2015, reddened Arctic Ocean sea ice and clouds have been observed, 13 minutes after totality. Gedzelman (2020) use a simple radiative transfer model to suggest that these yellow and orange colors observed from space are mainly caused by solar limb darkening. Figure 7 indeed shows that on 26 December 2019, the spectra in uncorrected satellite measurements could redden for $f_o > 0.33$. Because this reddening is described by the wavelength dependence of $f_o$, the reddening is automatically solved for with the solar irradiance correction of this paper and can therefore not be detected in a corrected product. Hence, the solar irradiance correction of this paper could be used to potentially prove that the yellow and orange colors in satellite images are indeed caused by solar limb darkening.

The solar irradiance correction is, besides the assumptions about the unpolarized state of $E_o$ and ignorance of 3-D effects, limited by the performance of the measurement instrument. For this paper, all TROPOMI TOA reflectance measurements had a SNR larger than 50. Measurement errors, 3-D effects and polarization of sunlight are expected to only play a role closer to the (ant)umbra and/or in the antumbra, and therefore did not leave signatures in the results of this paper. For potential applications of the solar irradiance correction to these regions in the future, it is advised to compare the calculated $f_o$ to the observed $f_o$ as in Sect. 3.1.2, which can help distinguishing between those artefacts and real air quality measurements.

The solar irradiance correction presented in this paper is a correction of the TOA reflectance spectrum. We have shown that the AAI successfully can be restored with the corrected TOA reflectances. Theoretically, any other product that is derived from TOA reflectance spectrum can be restored. The AAI is based on a ratio of absolute TOA reflectances in the UV which are directly affected by the eclipse obscuration. Differential spectral features are not expected to be directly affected by the eclipse obscuration. Therefore, we speculate that the solar irradiance correction could certainly also work for products that are based on differential spectral features, such as ozone, nitrogen dioxide and sulfur dioxide. However, high spectral resolution solar spectrum features that are not captured by the solar limb darkening measurements may have to be taken into account in the retrieval. As the photochemical activity in the Earth's atmosphere is driven by the TOA irradiation, solar eclipse related changes in the concentration of these gases could potentially be studied from space.

## 5   Summary and conclusions

In this paper, we presented a method to restore the TOA reflectance spectra in the penumbra and antumbra during solar eclipses, by computing the eclipse obscuration fraction as a function of location and time, fully taking into account wavelength-

dependent solar limb darkening. We applied the correction to UV TOA reflectances measured by TROPOMI in the penumbra during the annular solar eclipses on 26 December 2019 and 21 June 2020. We showed that the dark shade in the TOA reflectance maps for 380 nm, at the location of the Moon shadow, disappeared after the correction. For the eclipse on 26 December, we compared the calculated obscuration fractions to the estimated obscuration fractions at the ground pixels using measurements of the previous orbit and found a close agreement. Not taking into account solar limb darkening, however, resulted on 26 December 2019 in a mean underestimation of the obscuration fraction $f_o$ at 380 nm of 0.053 at disk center separations $X < 0.5$, and a maximum underestimation of 0.06.

The UV Absorbing Aerosol Index (AAI) is an air quality product derived from the TOA reflectance spectra. If no correction is applied, a significant increase of the TROPOMI AAI is measured in the shadow of the Moon. We explain this anomaly by the decreased measured TOA reflectance at 380 nm, which is used to define the Lambertian surface albedo in the model reflectance computations and propagates in the AAI formulae into a more 'blue' model UV spectrum, resulting in an increased AAI. That is, the AAI increase in the Moon shadow is not caused by a 'redder' measured UV spectrum.

With the restored TOA reflectance spectra, we computed a corrected version of the TROPOMI AAI on 26 December 2019 and 21 June 2020. For both eclipses, the AAI anomaly in the shadow of the Moon disappeared after the correction. For the eclipse on 26 December 2019, we showed that not taking into account solar limb darkening, however, could still result in an AAI overestimation of 6.7 points. We conclude that solar limb darkening cannot be neglected if the artificial Moon shadow signatures are to be removed.

For the eclipse on 21 June 2020, we found an AAI increase of ~1.5, as compared to its surrounding regions, in the restored TROPOMI product in Northwest China. We verified this AAI increase with AAI measurements by the GOME-2C satellite instrument on the same day but outside the Moon shadow. We attribute this restored AAI feature to the surface of the Taklamakan Desert and, possibly, desert dust aerosol. In this paper, we did not find an indication of absorbing aerosol changes in the Moon shadow (e.g. which are spatially correlated with the recent eclipse ground track). We conclude that the restored AAI product successfully can be used to detect real AAI rising phenomena.

The antumbra was not captured by the TROPOMI orbits during the annular solar eclipses studied in this paper, and the maximum $f_o$ was 0.92. Therefore, measurement errors, light travelling horizontally through the atmosphere between adjacent ground pixels and polarization of sunlight did not leave signatures in the corrected products, but their effect should carefully be reconsidered when restoring measurements in areas where $f_o > 0.92$ in the future.

We have demonstrated that the restored TOA reflectances during solar eclipses can be applied successfully to derive the AAI product. Since the method we developed has taken into account the wavelength dependence of the solar limb darkening, the method is applicable to the measured reflectances or radiances at all TROPOMI wavelengths. A solar eclipse flag is already included in the TROPOMI Level 1B product. With the addition of the obscuration fraction in the Level 1B product, all TROPOMI Level 2 products will benefit from the restored TOA reflectances or radiances. In principle, the method can also be applied to GOME-2, Sentinel-4/5 and other satellite instruments which measure the back-scattered and reflected solar radiation.

## Appendix A: Eclipse geometry at the ground pixel

In this appendix, we provide the method to compute the apparent lunar disk radius $r_\mathrm{m}$ and separation between the lunar and solar disk $X$ as experienced at the location of a ground pixel, from the geodetic coordinates $(\delta, \vartheta, h)$ and measurement time $t_1$ of that ground pixel.

The latitude $\delta$ is defined w.r.t. the Earth's equatorial plane and the longitude $\vartheta$ is defined w.r.t. the Greenwich meridian. Height $h$ is defined w.r.t. the Earth reference ellipsoid. The transformation of the ground pixel's geodetic coordinates $(\delta, \vartheta, h)$ to Cartesian coordinates in the geocentric Earth-fixed reference plane $(x_\mathrm{c}, y_\mathrm{c}, z_\mathrm{c})$ is given by

$$x_\mathrm{c} = (N_\delta + h)\cos\delta\cos\vartheta, \tag{A1}$$
$$y_\mathrm{c} = (N_\delta + h)\cos\delta\sin\vartheta, \tag{A2}$$
$$z_\mathrm{c} = ((1 - e^2)N_\delta + h)\sin\delta, \tag{A3}$$

with

$$e = \sqrt{2f - f^2}, \tag{A4}$$
$$N_\delta = \frac{a}{\sqrt{1 - e^2\sin^2\delta}}, \tag{A5}$$

where $a = 6378137$ m is the equatorial radius of the Earth and $f = 1/298.257223563$ is the flattening parameter of the Earth reference ellipsoid.

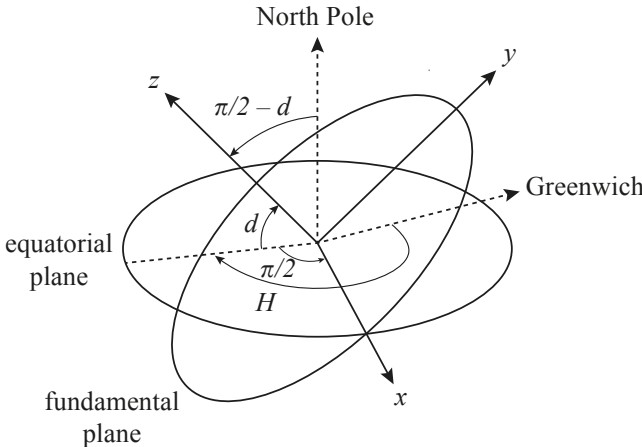

**Figure A1.** Transformation of the Earth-fixed coordinates to coordinates in the fundamental reference frame. Take the Earth-fixed $x$-axis pointed at Greenwich and rotate it around the $z$-axis (toward the North Pole) to $-(H - \pi/2)$. Then, rotate around the new $x$-axis by $\pi/2 - d$ to make the new $z$-axis parallel to the shadow axis (cf. Fig. 8).

The fundamental reference frame is a geocentric Cartesian coordinate system defined w.r.t. the shadow axis, which is the axis through the centers of the Moon and the Sun, as illustrated in Fig. 8. The $z$-axis of the fundamental reference frame originates in the Earth's center of mass and is parallel to the shadow axis. The $x$-axis is located in the equatorial plane and is positive toward the East. The $y$-axis completes the positive right-handed Coordinate system and is positive toward the North. The $xy$-plane of the fundamental reference frame ($z = 0$) is called the fundamental plane.

The orientation of the fundamental reference frame with respect to the Earth-fixed reference frame is defined by the shadow axis declination angle $d$ and the shadow axis Greenwich hour angle $H$ (see Fig. A1). The transformation of the Cartesian coordinates in the Earth-fixed reference plane to the Cartesian coordinates in the fundamental reference frame is computed as (Seidelmann, 1992, Eq. 8.331-3):

$$x_{\mathrm{f}} = \frac{1}{a}\left(x_{\mathrm{c}}\sin H + y_{\mathrm{c}}\cos H\right), \tag{A6}$$

$$y_{\mathrm{f}} = \frac{1}{a}\left(-x_{\mathrm{c}}\sin d\cos H + y_{\mathrm{c}}\sin d\sin H + z_{\mathrm{c}}\cos d\right), \tag{A7}$$

$$z_{\mathrm{f}} = \frac{1}{a}\left(x_{\mathrm{c}}\cos d\cos H - y_{\mathrm{c}}\cos d\sin H + z_{\mathrm{c}}\sin d\right). \tag{A8}$$

The declination $d$ is one of the Besselian elements, published by NASA[14] (Espenak and Meeus, 2006). A Besselian element is published as coefficients of a 3$^{\mathrm{rd}}$ order polynomial $B$ of time:

$$B = \sum_{N=0}^{3} c_n t^n, \tag{A9}$$

where time $t$ in Terrestrial Dynamical Time (TDT) is the measurement time $t_1$ in decimal hours with respect to a reference time $t_0$ commonly chosen close to the instant of greatest eclipse. Because $t_1$ by TROPOMI is stored as UTC, $t$ is computed as

$$t = t_1 + \frac{\Delta T}{3600} - t_0, \tag{A10}$$

where $\Delta T = \mathrm{TDT} - \mathrm{UTC}$ is in seconds. The values of $\Delta T$ and $t_0$ are published together with the Besselian elements for each eclipse. Another Besselian elements is the ephemeride hour angle $M$. Angle $H$ is computed as (Meeus, 1989)

$$H = M - \frac{360}{23 \times 3600 + 56 \times 60 + 4.098904}\Delta T, \tag{A11}$$

where $H$ and $M$ are in degrees. Note that all angular Besselian elements are published in degrees, and all dimensional Besselian elements are published per Earth's equatorial radius, $a$.

---

[14]The Besselian elements data can be retrieved from https://eclipse.gsfc.nasa.gov/SEcat5/SE2001-2100.html, by clicking on the gamma value.

The Besselian elements $l_1$ and $l_2$ are the radii of the penumbral and (ant)umbral shadow circle on the fundamental plane, respectively, as illustrated in Fig. A2. The vertex angles $f_1$ and $f_2$ of the penumbral and (ant)umbral shadow, respectively, are also published as Besellian elements. Note that $l_2$ is positive for annular eclipes and negative for total eclipses, while $l_1$ is always positive.

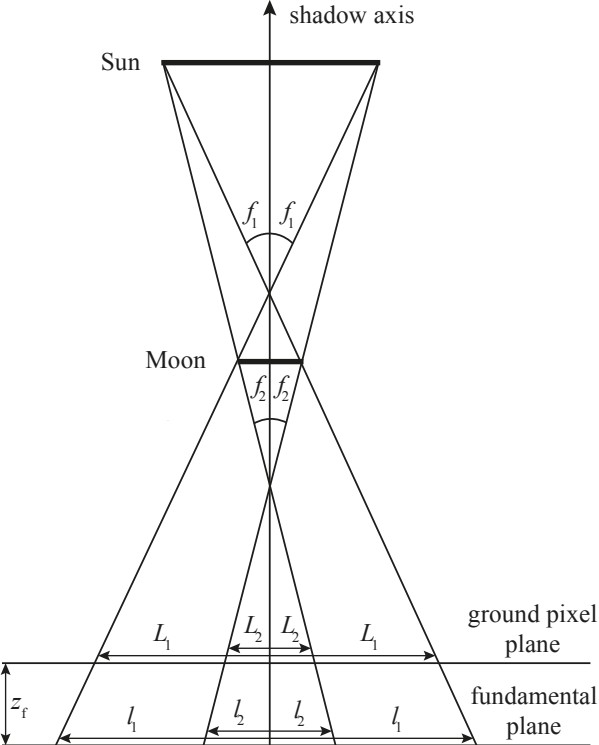

**Figure A2.** Definition of $l_1$, $l_2$, $L_1$, $L_2$, $f_1$ and $f_2$.

The coordinates of the shadow axis in the fundamental reference frame, $x_{\mathrm{sdw}}$, $y_{\mathrm{sdw}}$ and $z_{\mathrm{sdw}}$ are also published as Besselian elements. The distance $m$ from the ground pixel to the shadow axis, parallel to the fundamental plane, is then

$$m = \sqrt{(x_{\mathrm{f}} - x_{\mathrm{sdw}})^2 + (y_{\mathrm{f}} - y_{\mathrm{sdw}})^2}. \tag{A12}$$

The plane parallel to the fundamental plane through the ground pixel is the so-called ground pixel plane. The radii of the penumbra and the (ant)umbra on the ground pixel plane, $L_1$ and $L_2$ respectively, can readily be computed by

$$L_1 = l_1 - z_{\mathrm{f}} \tan(f_1), \tag{A13}$$
$$L_2 = l_2 - z_{\mathrm{f}} \tan(f_2). \tag{A14}$$

If $z_f > 0$, a ground pixel can be eclipsed. One can compute whether a ground pixel is eclipsed and what shadow type is experienced by comparing $m$ to $L_1$ and $L_2$. Quantities $r_m$ and $X$ can be expressed in terms of $L_1$, $L_2$ and $m$, as

$$r_m = \frac{L_1 - L_2}{L_1 + L_2},$$

(A15)

$$X = \frac{2m}{L_1 + L_2}.$$

(A16)

For the derivations of Eq. A15 and A16, we refer to Sect. 8.3623 of Seidelmann (1992).

## Appendix B: Error propagation

In this appendix, we show the effect of the solar irradiance correction on the precision of the AAI. It should be recalled from Eq. 3 that the corrected measured TOA reflectance, $R^{int}(\lambda)$, is computed from the measured TOA reflectance by TROPOMI, $R^{meas}(\lambda)$, and the calculated obscuration fraction, $f_o(\lambda)$. We assume that the noise of $R^{meas}(\lambda)$ and the noise of $f_o(\lambda)$ are normally distributed with standard deviations $\sigma_{R^{meas}(\lambda)}$ and $\sigma_{f_o(\lambda)}$, respectively. Also, we assume that the noise of $R^{meas}(\lambda)$ is not correlated with the noise of $f_o(\lambda)$. Then, we may compute the precision of $R^{int}(\lambda)$ as follows:

$$\sigma_{R^{int}} = R^{int} \cdot \sqrt{\left(\frac{\sigma_{R^{meas}}}{R^{meas}}\right)^2 + \left(\frac{\sigma_{f_o}}{1 - f_o}\right)^2}.$$

(B1)

$\sigma_{R^{meas}(\lambda)}$ is provided in the current operational TROPOMI L2 AAI product for $\lambda = 340$ nm and $\lambda = 380$ nm. $\sigma_{f_o(\lambda)}$ depends on the precision of the geometrical eclipse prediction, i.e. $\alpha$ in Eq. 9, and the precision of the solar limb darkening function, $\Gamma$. The geometrical eclipse prediction was verified with the predictions by NASA (see Sect. 2.4). The largest source of uncertainty for $\alpha$ in the present era (1800 CE to present) is the Moon's surface topography, which causes the lunar disk circumference to deviate from a perfect circle.[15] Our and NASA's eclipse predictions do not include these effects of mountains and valleys along the edge of the Moon, which may shift the limits of the eclipse path north or south by $\sim 1$ to 3 kilometers, and may change the eclipse duration by $\sim 1$ to 3 seconds.[16] For the solar eclipse of 21 June 2020, we added a time increment of 3 seconds to estimate the effect of a local eclipse timing error due to the Moon's topography on $f_o$ and the AAI. At the ground pixels for which $f_o > 0$, the average absolute changes in $f_o$ and the AAI were 0.00049 and 0.00588, respectively. Hence, in what follows in this appendix, we assume that there is no error in $\alpha$.

Pierce and Slaughter (1977) provide the probable error, $Pe$, of the estimated solar limb darkening function $\Gamma(\lambda, r)$ for the tabulated set of $\lambda$'s, which is independent of $r$. We assume that the measurement noise of $\Gamma$ was normally distributed with standard deviation $\sigma_\Gamma(\lambda) = Pe(\lambda)/0.6745$. We also assume that the noise of $\Gamma$ was uncorrelated in $r$-space. We performed a

---

[15]See https://eclipse.gsfc.nasa.gov/SEhelp/limb.html, visited on 27 March 2021.

[16]See https://eclipse.gsfc.nasa.gov/SEpath/SEpath2001/SE2020Jun21Apath.html, visited on 27 March 2021.

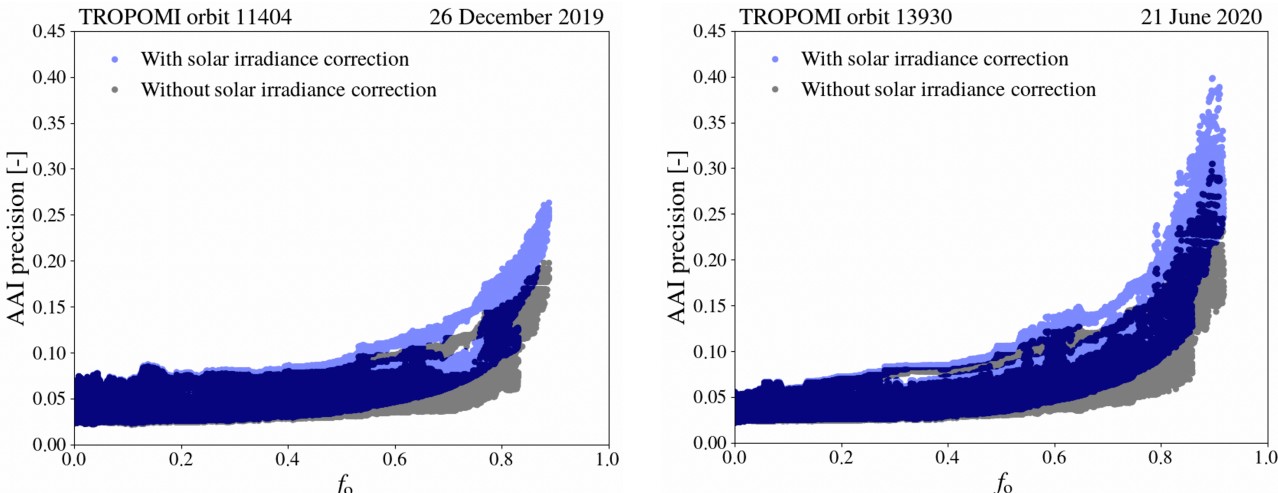

**Figure B1.** Precision of the AAI in the eclipsed TROPOMI orbits on 26 December 2019 (left) and 21 June 2020 (right) as a function of obscuration fraction $f_o$ at 380 nm, with solar irradiance correction such that both $\sigma_{R^{\mathrm{meas}}}$ and $\sigma_\Gamma$ are propagated (semi-transparent blue dots) and without solar irradiance correction such that only $\sigma_{R^{\mathrm{meas}}}$ is propagated (semi-transparent black dots).

Monte Carlo error propagation simulation to estimate the error of the obscuration fraction, $\sigma_{f_o}(\lambda)$. That is, to each $\Gamma(\lambda, r)$ at
$\lambda = 340$ nm and 380 nm, we added 100 times a randomly generated normally distributed error $\sigma_\Gamma(\lambda)$, repeated the computation of $f_o$ (Eq. 9) for each sample, and computed $\sigma_{f_o}$ as the standard deviation of $f_o$. The precision of the AAI can be computed as follows:

$$\sigma_{\mathrm{AAI}} = 100 \sqrt{\left( \frac{\sigma_{R^{\mathrm{model}}_{340}}}{\ln(10) R^{\mathrm{model}}_{340}} \right)^2 + \left( \frac{\sigma_{R^{\mathrm{int}}_{340}}}{\ln(10) R^{\mathrm{int}}_{340}} \right)^2 }. \tag{B2}$$

The expression for the precision of the modeled reflectance at 340 nm, $\sigma_{R^{\mathrm{model}}_{340}}$, as a function of $\sigma_{R^{\mathrm{int}}_{380}}$, can readily be derived by analytically solving $\sigma_{R^{\mathrm{model}}_{340}} = \sigma_{A_s} |\partial R^{\mathrm{model}}_{340} / \partial A_s|$ and $\sigma_{A_s} = \sigma_{R^{\mathrm{int}}_{380}} |\partial A_s / \partial R^{\mathrm{int}}_{380}|$. Possible calibration (offset) errors of TROPOMI and model uncertainties of DAK are excluded from this analysis.

Figure B1 shows $\sigma_{\mathrm{AAI}}$ at the ground pixels of the eclipsed TROPOMI orbits during the solar eclipses that were discussed in this paper. The results are presented as a function of obscuration fraction $f_o$ at 380 nm, with solar irradiance correction applied (semi-transparent blue dots) and without solar irradiance correction applied (semi-transparent black dots). In the absence of a solar eclipse ($f_o = 0$), $\sigma_{\mathrm{AAI}}$ for with and without solar irradiance correction is identical, since $R^{\mathrm{int}} = R^{\mathrm{meas}}$ and $\sigma_{R^{\mathrm{int}}} = \sigma_{R^{\mathrm{meas}}}$ (see Eq. B1). In the presence of a solar eclipse, $\sigma_{\mathrm{AAI}}$ increases with increasing $f_o$. When no solar irradiance correction is applied, again $R^{\mathrm{int}} = R^{\mathrm{meas}}$, but $R^{\mathrm{int}}$ is decreased at both 340 and 380 nm in the Moon shadow, which increases $\sigma_{\mathrm{AAI}}$ through the division by $R^{\mathrm{int}}_{340}$ and through the increased $\sigma_{A_s}$ and $\sigma_{R^{\mathrm{model}}_{340}}$ (Eq. B2). When a solar irradiance correction is applied, $R^{\mathrm{int}} > R^{\mathrm{meas}}$ and is not decreased anymore by the eclipse. However, the additional solar limb darkening noise term in Eq. B1 increases $\sigma_{\mathrm{AAI}}$, which is most significant at relatively large $f_o$ resulting from the division by $1 - f_o$. The maximum AAI precisions, after a

solar irradiance correction in the TROPOMI orbits on 26 December 2019 and 21 June 2016, are 0.26 and 0.40 respectively. The precisions in the uncorrected case are respectively $\sim 0.05$ and $\sim 0.10$ better, but the uncorrected AAI value itself is off by many points as shown in for example Fig. 17.

*Author contributions.* P.S. came up with the idea to analyze solar eclipses. V.T. came up with the idea to correct eclipsed satellite data and performed all computations. P.W. weekly commented on the intermediate results and guided V.T. to focus on the most relevant aspects. V.T. wrote the manuscript. All authors read the manuscript, provided feedback that lead to improvements and were involved in the selection of the results presented in this paper.

*Competing interests.* The authors declare no competing interests.

*Acknowledgements.* This work is part of the research programme User Support Programme Space Research (GO) with project number ALWGO.2018.016, which is (partly) financed by the Dutch Research Council (NWO). We thank Maarten Sneep for the help with the TROPOMI AAI algorithm and we thank Olaf Tuinder for preparing the GOME-2C data. We thank the anonymous reviewers for their constructive comments.

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
