# Peer review of "Restoring the top-of-atmosphere reflectance during solar eclipses: a proof of concept with the UV Absorbing Aerosol Index measured by TROPOMI"

_Atmospheric Chemistry and Physics, 2020_

## Referee Comment (RC1) · Anonymous Referee #2 · 29 Dec 2020

The paper describes a method to correct TROPOMI/S5P observations during solar eclipses. The shadow of the moon reduces the incident irradiance. In the derivation of reflectances from this observations the irradiance of non-eclipse conditions is used, therefore these reflectances are wrong and retrieval algorithms using these reflectances yield wrong results. Therefore observations during eclipses are currently not used for further analysis. The observations can be corrected quite easily by using the reduced incident irradiance to derive the reflectance. Consistently with other studies, it is shown that in order to compute the reduced irradiance it is important to take

into account the solar limb darkening. The authors derived such a correction method and apply it to the derivation of the aerosol absorption index. Using the corrected reflectances they obtain reasonable results also during the eclipse which are consisted with observations derived in non-eclipse conditions. Satellite based aerosol and trace gas measurements my reveal interesting effects of the solar eclipse on the composition of the atmosphere, however this is not investigated in the study. The paper is generally well written in good English and the number of figures is appropriate. I recommend publication in ACP after some revisions as suggested in my comments below.

General comments:

- In the paper the method to correct observations during solar eclipses is described. It is mentioned in the introduction that corrected observations can be used to study effects of the solar eclipse on atmospheric composition. I suggest to include such a study, this would increase the scientific relevance of the paper significantly.

- Motivate, why it is interesting to study solar eclipses and their effects on atmospheric composition. In the abstract it is written that it is "may be of particular interest", this sounds as if the authors do not know themselves whether it is really interesting ...

- How important is this correction method? How frequently are the observations disturbed by solar eclipses?

Specific comments:

l.1 "Solar eclipses reduce the measured top-of-atmosphere (TOA) reflectances as derived by Earth observation satellites, because the solar irradiance that is used to compute these reflectances is commonly measured before the start of the eclipse." -> This sentence in the beginning is a little confusing, rephrase? First mention that solar irradiance is reduced in moon shadow. Then mention, that normalized quantity "reflectance" should not be affected when reduced irradiance is used for normalization and write that this is not yet done in the operational processing of the data ...

l.12 "in a maximum Moon shadow signature in the AAI of 6.7 points increase" -> what is a "point"?

l.206: "We provide the recipe for the computation of X ... "-> Have you compared your derivation to that presented in Ockenfuss et al. 2020?

l.287: "The maximum underestimation of f0 at 380 nm, when using $\Gamma = 1$, was 0.06 at 6.04°N latitude and 107.19°E longitude." -> What is the maximum underestimation when limb darkening is taken into account ...

l.318: "The negative mean AAI are partly caused by the scattering of cloud droplets, but also due to a radiometric calibration offset and degradation in the TROPOMI irradiance data ..." -> Please explain: 1. Why is AII negative for cloud scattering, 2. Why is there a radiometric calibration offset, 3. Why is there a degradation in the TROPOMI irradiance data

Fig.16: I have a general question about the interpretation AII. It seems that in the figure most higher values of AII are not due to aerosols but due to clouds and sunglint? Also values seem to be higher towards the edges of the orbit, are these AII values correct? Can you indicate an area in the figure which clearly shows an increased AII due to the presence of aerosol?

l.385: "at 36°-42°N latitude and 78°-86°E longitude" -> could you mark this region in Fig.20?

l.440: "Hence, the solar irradiance correction of this paper could be used to potentially prove that the yellow and orange colors in satellite images are indeed caused by solar limb darkening." -> Can you try this and include a corrected image? This should not be much work?

Technical corrections:

l.17: "can be used to detect real AAI rising phenomena ... " -> "can be used to detect real AAI rising phenomena during a solar eclipse ... "

l.90: "and on phi-phi0" -> mention that if 3D effects matter the absolute azimuth angles need to be taken into account

---

## Referee Comment (RC2) · Anonymous Referee #1 · 12 Jan 2021

General comments:

The manuscript by Trees et al. describes a technique to correct for the change in the top-of-atmosphere (TOA) solar spectral irradiance during a partial or annual solar eclipse. The technique is based on earlier works by Koepke et al. (2001), Bernhard and Petkov (2019) and Ockenfuß et al. (2020). However, the authors generalized these calculations by also including the case of an annular eclipse (The formulas presented by Koepke et al. (2001) only consider partial and total eclipses). The method makes satellite measurements collected during a solar eclipse available for the retrieval of

properties of the Earth's atmosphere. The technique is sound and the validation results show convincingly that it is working. The topic is suitable for Atmospheric Chemistry and Physics.

My main issue is that a solar eclipse is a very rare event and applicability of the method is therefore limited. If data from one or two satellite orbits have to be discarded because of contamination by the Moon's shadow, the data loss is minor and interpolations using data from adjacent orbits should satisfy most needs. The more interesting question is whether the technique is accurate enough to detect changes in atmospheric properties during an eclipse and could help to evaluate atmospheric processes initiated by an eclipse. For example, there could potentially be a change in aerosol properties during the eclipse because aerosol hygroscopic growth is likely affected by the reduced air temperature (and the resulting effect on relative humidity) during an eclipse. For the correction method described by the authors to be useful, the uncertainty of the correction must be smaller than the expected change in aerosol properties induced by an eclipse. What is the evidence that the uncertainty is indeed sufficiently small? The authors should try to estimate the uncertainty of their correction as it applies to the ultraviolet (UV) Absorbing Aerosol Index (AAI). This would further demonstrate the strength of their method and increase the scientific relevance of the paper.

The authors chose to validate their method by comparing retrievals of the UV AAI with and without correction for the Moon's shadow. The UV AAI is a hard-to-interpret indicator of aerosol absorption properties. An alternative metric would be the aerosol absorption optical depth (AAOD), which is similarly defined as the widely-used aerosol optical depth (AOD), except that optical depth refers to the absorbing part of aerosols only and not to the extinction (from absorption and scattering), as it is the case for the AOD. Hence the AAOD is a more useful quantity to describe aerosol absorption properties than the AAI. It would be helpful if the authors could briefly explain how the AAI relates to the AAOD and/or provide a reference.

Specific comments

The abstract should be improved for clarity. Technical terms that are not commonly used should be avoided or defined. For example:

– L7: "eclipse obscuration fraction" is not a commonly used term. While I don't object to its use after it is properly defined, it might be better to either avoid this term in the abstract, or provide a short verbal definition.

– L9: The sentence "We verify the calculated obscuration with the observed obscuration using an uneclipsed orbit." is misleading. The paper compares *data products* obtained with and without obscurations, not obscurations per se.

– L12: The sentence "...would result in [...] in a maximum Moon shadow signature in the AAI of 6.7 points increase." is difficult to understand without reading the paper first and should either be reworded or deleted.

L54: Regarding: "Such wavelength-independent approximations of the eclipse obscuration fraction based on the the overlapping disks indeed could work well to estimate the shortwave fluxes." "works well" should be quantified. Weather an approximation "works well" depends on the desired accuracy. Also, the word "the" before "overlapping" is repeated.

L90: Regarding "I depends on mu = cos(theta) where theta is the viewing zenith angle," This is misleading as it could be interpreted that I(theta) = I (0) * cos(theta). If the Earth were a Lambertian Reflector, the radiance would be independent of the viewing angle theta (i.e.: I(theta) = I(0)). Since the Earth is not a Lambertian Reflector, the radiance *will* depend on the viewing angle. While this dependency could be expressed as a *function* of cos(theta), this is not clear from the sentence. I suggest to just mention that I depends on the viewing angle without mentioning cos(theta).

L94: " The fraction of sunlight that is blocked by the Moon is the eclipse obscuration fraction, f_0." This is not a good definition as "sunlight" is not a physical quantity. Furthermore, because of solar limb darkening, f_0 is not defined by the geometric

area of the solar disk that is blocked by the moon, but depends on wavelength – see for example Figure 7. This should already be mentioned here. To emphasize the wavelength dependence, the symbol f_0(lambda) should be used instead of f_0. So I would say:

"The fraction of the TOA spectral irradiance E_0(lambda) that is blocked by the Moon is the wavelength-dependent eclipse obscuration fraction, f_0(lambda). The remaining solar spectral irradiance at TOA is (1-f_0(lambda))E_0(lambda)." In general, it would be helpful to add "(lambda)" after all spectral quantities that depend on wavelength.

Eqs. (6) and (7). Please replace E_0 and f_0 with E_0(lambda) and f_0(lambda) to emphasize that these are spectral quantities like I(lambda). See also Figure 7.

L290: Before explaining how to interpret the AAI, its definition (i.e., Eq. (16)) should be presented and explained.

L292: While it may be possible to calculate the AAI in the present of clouds, is the result of any value? Absorbing aerosols are typically close to the surface (at least in the vicinity of urban centers) and cannot be "seen" by a satellite below a moderately thick cloud.

Adding to my general comment, it is beyond the scope of the paper to discuss the value of the AAI to characterize aerosol absorption. Still, the authors should better explain why they chose this parameter to validate their correction method. For example, Eq. (16) could lead to a AAI different from 0 for the case of non-absorbing small aerosol particles. Hence the AAI could potentially indicate absorbing aerosols when in fact non-absorbing aerosol was present.

Figure 15: The scatter is rather large. So the figure's value to validate the correction method is rather limited for X > 0.7. This could be mentioned.

Technical comments

L41: "have been taken," > "have been observed,"

L51: Why "instead"? The verb "approximated" already implies that this is a simplification.

L168: low > small

L241: For clarity, please explain "scanline". (E.g., the line at Earth's surface defined by the satellite swath that is roughly oriented East-West)

L275: "Sect.." > "section."

L317L Moon > Moon's

L331: 110405 > 11405

L358: delete "still"

L415: What is "chord"?

---

## Author Comment (AC1) · 27 Mar 2021

**Response to comment of Anonymous Referee #1 on "Restoring the top-of-atmosphere reflectance during solar eclipses: a proof of concept with the UV Absorbing Aerosol Index measured by TROPOMI" by Victor Trees et al.**

Victor Trees[1,2], Ping Wang[1], and Piet Stammes[1]

[1]Royal Netherlands Meteorological Institute (KNMI), De Bilt, The Netherlands
[2]Delft University of Technology, Delft, The Netherlands

**Correspondence:** Victor Trees (victor.trees@knmi.nl)

We thank the reviewer for his/her careful reading and for the comments and suggestions, which have improved the manuscript. Below, we give in *black italic* the reviewer's comment, in black our response, and in red the changed text in the manuscript.

*General comments:*

*The manuscript by Trees et al. describes a technique to correct for the change in the top-of-atmosphere (TOA) solar spectral irradiance during a partial or annual solar eclipse. The technique is based on earlier works by Koepke et al. (2001), Bernhard and Petkov (2019) and Ockenfuß et al. (2020). However, the authors generalized these calculations by also including the case of an annular eclipse (The formulas presented by Koepke et al. (2001) only consider partial and total eclipses). The*

10 *method makes satellite measurements collected during a solar eclipse available for the retrieval of properties of the Earth's atmosphere. The technique is sound and the validation results show convincingly that it is working. The topic is suitable for Atmospheric Chemistry and Physics.*

*My main issue is that a solar eclipse is a very rare event and applicability of the method is therefore limited. If data from one*
15 *or two satellite orbits have to be discarded because of contamination by the Moon's shadow, the data loss is minor and interpolations using data from adjacent orbits should satisfy most needs. The more interesting question is whether the technique is accurate enough to detect changes in atmospheric properties during an eclipse and could help to evaluate atmospheric processes initiated by an eclipse. For example, there could potentially be a change in aerosol properties during the eclipse because aerosol hygroscopic growth is likely affected by the reduced air temperature (and the resulting effect on relative hu-*
20 *midity) during an eclipse. For the correction method described by the authors to be useful, the uncertainty of the correction must be smaller than the expected change in aerosol properties induced by an eclipse. What is the evidence that the uncertainty is indeed sufficiently small? The authors should try to estimate the uncertainty of their correction as it applies to the ultraviolet (UV) Absorbing Aerosol Index (AAI). This would further demonstrate the strength of their method and increase the scientific*

*relevance of the paper.*

25

We thank the reviewer for her/his critical comment on the applicability and accuracy of the eclipse correction method. We address the points mentioned in the above comment separately:

- *My main issue is that a solar eclipse is a very rare event and applicability of the method is therefore limited. If data from one or two satellite orbits have to be discarded because of contamination by the Moon's shadow, the data loss is minor*

30

- *and interpolations using data from adjacent orbits should satisfy most needs.*

Every year, there are about 2 to 3 solar eclipses (with a yearly average of 2.4 solar eclipses)[1]. Since the start of the nominal operational mode of TROPOMI in May 2018, 7 solar eclipses occurred, 6 of which have been measured by TROPOMI. In the paper, in the second paragraph of the Introduction, we mention that, sometimes, such eclipse anomalies propagate into anomalies in temporal average maps without raising an eclipse flag, potentially resulting in false conclusions about the

35

mean aerosol effect in that time period. An example of a *monthly average* AAI map of the GOME-2 satellite instrument that is distorted by a solar eclipse can be found on https://d1qb6yzwaaq4he.cloudfront.net/airpollution/absaai/GOME2B/ monthly/images/2019/GOME-2B_AAI_map_201912.png. The formulae in the Appendix of this paper can also be used to only compute an eclipse flag. Indeed, an eclipse flag computation is not new and the eclipse anomalies could be omitted by raising an eclipse flag and discarding the data, however, the advantage of a correction is that data is not

40

lost and can potentially be used to study solar eclipses from space. In addition, the successful correction shows that the physics of the shadow is understood.

We have added the following sentence to the second paragraph of the Introduction:

l.28: "Since the start of the nominal operational mode of the TROPOMI spectrometer instrument on board the S5P satellite in May 2018, 7 solar eclipses occurred, 6 of which have been measured by TROPOMI."

45

We have changed the last sentence in the second paragraph of the Introduction:

l.36: "Sometimes, such eclipse anomalies propagate into anomalies in temporal average maps without raising an eclipse flag, potentially resulting in false conclusions about the mean aerosol effect in that time period." -> "TROPOMI data contains an eclipse flag indicating the eclipse occurrence at a ground pixel. For satellite instruments that do not contain an eclipse flag, such as the GOME-2 instrument, these eclipse anomalies propagate into anomalies in temporal average

50

maps, potentially resulting in false conclusions about the mean aerosol effect in that time period."

Also, we have added the following footnote to the second paragraph of the Introduction:

l.40: An example of a *monthly average* AAI map of the GOME-2 satellite instrument that is distorted by a solar eclipse can be found on https://d1qb6yzwaaq4he.cloudfront.net/airpollution/absaai/GOME2B/monthly/images/2019/GOME-2B_ AAI_map_201912.png, visited on 22 February 2021.
* * *
[1]https://eclipse.gsfc.nasa.gov/SEpubs/5MCSE.html

55       – *The more interesting question is whether the technique is accurate enough to detect changes in atmospheric properties during an eclipse and could help to evaluate atmospheric processes initiated by an eclipse. For example, there could potentially be a change in aerosol properties dur- ing the eclipse because aerosol hygroscopic growth is likely affected by the reduced air temperature (and the resulting effect on relative humidity) during an eclipse.*

Evaluating atmospheric processes initiated by the eclipse would, besides the measurement and correction accuracy, also

60     require an estimation of the 'non-eclipse state', i.e. the situation if no eclipse would have occurred, which is unknown. However, if changes are detected that are spatially correlated with the recent eclipse ground track, atmospheric processes initiated by an eclipse could potentially be identified. In this paper, we did not find an indication of a change of the absorbing aerosol effect that was spatially correlated with the recent eclipse ground track. Montornès et al. (2016) modeled a local surface temperature response of $\sim$-1K to $\sim$-3K. We speculate that aerosol properties do not rapidly

65     change in a the few hours of reduced temperature during the eclipse. We speculate, however, that chemical processes requiring sunlight could possibly change in a few hours during the eclipse, such as NO2 or tropospheric ozone, but this should be investigated in a future study. We added the following sentence to the Abstract:

l.15: "No indication of local absorbing aerosol changes caused by the eclipses was found."

We added the following sentence to the Conclusion:

70     l.489: "In this paper, we did not find an indication of absorbing aerosol changes in the Moon shadow (e.g. which are spatially correlated with the recent eclipse ground track)."

      – *For the correction method described by the authors to be useful, the uncertainty of the correction must be smaller than the expected change in aerosol properties induced by an eclipse. What is the evidence that the uncertainty is indeed sufficiently small? The authors should try to estimate the uncertainty of their correction as it applies to the ultraviolet*

75     *(UV) Absorbing Aerosol Index (AAI). This would further demonstrate the strength of their method and increase the scientific relevance of the paper.*

We agree with the reviewer that a detection of an eclipse-initiated phenomenon only can be done when its signal is significantly stronger than the noise of the corrected product itself. For the two eclipse cases discussed in this paper, we did not find significant features in the corrected AAI product that could indicate eclipse-initiated phenomena. In the new

80     Appendix B of this document, which we also added to the paper, we analyze the effect of the solar irradiance correction on the AAI precision.

We also added the following sentences:

l.316: "In Appendix B we provide an analysis of the precision of the AAI during the solar eclipses studied in this paper."

85     l.394: "Note that this AAI change is larger than the maximum standard AAI error in orbit 13930 of 0.40 (see Appendix B)."

*The authors chose to validate their method by comparing retrievals of the UV AAI with and without correction for the Moon's shadow. The UV AAI is a hard-to-interpret indicator of aerosol absorption properties. An alternative metric would be the aerosol absorption optical depth (AAOD), which is similarly defined as the widely-used aerosol optical depth (AOD), except*
90  *that optical depth refers to the absorbing part of aerosols only and not to the extinction (from absorption and scattering), as it is the case for the AOD. Hence the AAOD is a more useful quantity to describe aerosol absorption properties than the AAI. It would be helpful if the authors could briefly explain how the AAI relates to the AAOD and/or provide a reference.*

The aerosol absorption optical depth is defined as AAOD = AOD (1- SSA) (Sun et al., 2019). The AAI depends on various
95  parameters such as the AOD, SSA and height (de Graaf et al., 2005). Therefore, the AAOD is not retrievable from AAI unless a value for SSA and height is assumed or taken from a model. We have added a sentence to the AAI description:

l.314: "In the next paragraph, we provide a brief introduction to the AAI. For more details about the sensitivity of the AAI to atmosphere and surface parameters, we refer to..." -> "In the next paragraph, we provide a brief introduction to the AAI. The
100  AAI depends on various parameters such as the aerosol optical depth (AOD), single scattering albedo (SSA) and aerosol layer height (ALH). For more details about the sensitivity of the AAI to atmosphere and surface parameters, we refer to..."

*Specific comments:*
*The abstract should be improved for clarity. Technical terms that are not commonly used should be avoided or defined. For*
105  *example:*

- *L7: "eclipse obscuration fraction" is not a commonly used term. While I don't object to its use after it is properly defined, it might be better to either avoid this term in the abstract, or provide a short verbal definition.*

  We changed the sentence:

  L7: "... how to compute the eclipse obscuration fraction ..." -> "... how to compute the obscuration during a solar eclipse
110  ..."

- *L9: The sentence "We verify the calculated obscuration with the observed obscuration using an uneclipsed orbit." is misleading. The paper compares \*data products\* obtained with and without obscurations, not obscurations per se.*

  We have changed the sentences as follows:

  l.10: "We verify the calculated obscuration with the observed obscuration using an uneclipsed orbit." -> "We compare
115  the calculated obscuration to the estimated obscuration using an uneclipsed orbit."

  l.76: "...and we show how the calculated obscuration fraction can be verified by using measurements in an uneclipsed orbit..." -> "...and we show how the calculated obscuration fraction can be compared to the estimated obscuration fraction from measurements in an uneclipsed orbit..."

l.219: "We use the example of 26 December 2019 to verify the calculated obscuration fractions..." -> "We use the example of 26 December 2019 to compare the calculated obscuration fractions to the estimated obscuration fractions from observations in an uneclipsed orbit..."

l.472: "...we compared the calculated obscuration fractions to the observed obscuration fractions at the ground pixels using measurements of the previous orbit... " -> "...we compared the calculated obscuration fractions to the estimated obscuration fractions at the ground pixels using measurements of the previous orbit..."

– *L12: The sentence ". . .would result in [. . .] in a maximum Moon shadow signature in the AAI of 6.7 points increase." is difficult to understand without reading the paper first and should either be reworded or deleted.*

We simplified this part in the abstract as follows:

L10: "In the corrected products, the signature of the Moon shadow disappeared. Not taking into account solar limb darkening, however, would result in a maximum underestimation of the obscuration fraction of 0.06 at 380 nm on 26 December 2019, and in a maximum Moon shadow signature in the AAI of 6.7 increase." -> "In the corrected products, the signature of the Moon shadow disappeared, but only if wavelength-dependent solar limb darkening is taken into account."

– *L54: Regarding: "Such wavelength-independent approximations of the eclipse obscuration fraction based on the the overlapping disks indeed could work well to estimate the shortwave fluxes." "works well" should be quantified. Weather an approximation "works well" depends on the desired accuracy. Also, the word "the" before "overlapping" is repeated.*

We agree that the succesfulness of such an approximation will depend on the accuracy of the results versus the desired accuracy, which explains our usage of the word "could" in this sentence. How accurate the results using such an approximation are, depends on the assumptions made. We did not study and so quantify the effect of neglecting the wavelength dependence of the obscuration on the accuracy of shortwave fluxes (i.e. integrated over the spectral domain), because we study the reflectances as functions of wavelength. We changed the sentence as follows:

L58: "Such wavelength-independent approximations of the eclipse obscuration fraction based on the the overlapping disks indeed could work well to estimate the shortwave fluxes." -> "Such wavelength-independent approximations of the eclipse obscuration fraction based on the the overlapping disks indeed could work well to estimate the shortwave fluxes, depending on the desired accuracy."

– *L90: Regarding "I depends on mu = cos(theta) where theta is the viewing zenith angle," This is misleading as it could be interpreted that I(theta) = I (0) * cos(theta). If the Earth were a Lambertian Reflector, the radiance would be independent of the viewing angle theta (i.e.: I(theta) = I(0)). Since the Earth is not a Lambertian Reflector, the radiance \*will\* depend on the viewing angle. While this dependency could be expressed as a \*function\* of cos(theta), this is not clear from the sentence. I suggest to just mention that I depends on the viewing angle without mentioning cos(theta).*

We changed the sentence as follows:

L95: "Also, $I$ depends on $\mu = \cos\theta$ where $\theta$ is the viewing zenith angle, on $\mu_0 = \cos\theta_0$ where $\theta_0$ is the solar zenith angle, on the viewing azimuth angle $\varphi$ and on the solar azimuth angle $\varphi_0$." -> "Also, $I$ depends on the viewing zenith angle $\theta$, the solar zenith angle $\theta_0$, the viewing azimuth angle $\varphi$ and the solar azimuth angle $\varphi_0$. Furthermore, we use the definitions $\mu = \cos\theta$ and $\mu_0 = \cos\theta_0$."

– *L94: " The fraction of sunlight that is blocked by the Moon is the eclipse obscuration fraction, $f_o$." This is not a good definition as "sunlight" is not a physical quantity. Furthermore, because of solar limb darkening, $f_o$ is not defined by the geometric area of the solar disk that is blocked by the moon, but depends on wavelength – see for example Figure 7. This should already be mentioned here. To emphasize the wavelength dependence, the symbol $f_o(\lambda)$ should be used instead of $f_o$. So I would say: "The fraction of the TOA spectral irradiance $E_o(\lambda)$ that is blocked by the Moon is the wavelength-dependent eclipse obscuration fraction, $f_o(\lambda)$. The remaining solar spectral irradiance at TOA is $(1-f_o(\lambda))E_o(\lambda)$." In general, it would be helpful to add "($\lambda$)" after all spectral quantities that depend on wavelength.*

We have changed the sentence as suggested:

L99: "The fraction of sunlight that is blocked by the Moon is the eclipse obscuration fraction, $f_\mathrm{o}$. The remaining solar irradiance at TOA is $(1-f_\mathrm{o})E_0$." -> "The fraction of the TOA spectral irradiance $E_0(\lambda)$ that is blocked by the Moon is the wavelength-dependent eclipse obscuration fraction, $f_\mathrm{o}(\lambda)$. The remaining spectral irradiance at TOA is $[1-f_\mathrm{o}(\lambda)]E_0(\lambda)$."

We added $(\lambda)$ to $f_\mathrm{o}$ and $E_0$ in Eqs. 1, 2, 3, 6, and 7, and in the text where relevant.

– *Eqs. (6) and (7). Please replace $E_0$ and $f_o$ with $E_0(\lambda)$ and $f_o(\lambda)$ to emphasize that these are spectral quantities like $I(\lambda)$. See also Figure 7.*

We added $(\lambda)$ to $f_\mathrm{o}$ and $E_0$ in Eqs. 1, 2, 3, 6, and 7, and in the text where relevant.

– *L290: Before explaining how to interpret the AAI, its definition (i.e., Eq. (16)) should be presented and explained.*

We thank the reviewer for this suggestion. We swapped the first and second paragraph of Section 3.1.3.

– *L292: While it may be possible to calculate the AAI in the present of clouds, is the result of any value? Absorbing aerosols are typically close to the surface (at least in the vicinity of urban centers) and cannot be "seen" by a satellite below a moderately thick cloud. Adding to my general comment, it is beyond the scope of the paper to discuss the value of the AAI to characterize aerosol absorption. Still, the authors should better explain why they chose this parameter to validate their correction method. For example, Eq. (16) could lead to a AAI different from 0 for the case of non-absorbing small aerosol particles. Hence the AAI could potentially indicate absorbing aerosols when in fact non-absorbing aerosol was present.*

The AAI can particularly indicate absorbing aerosols when they are located above the clouds. In that case, the AAI is equivalent to the AAI for an aerosol layer above a bright surface (see de Graaf et al., 2005). When the aerosol layer is

below the cloud, the AAI is indeed determined by the cloud scattering properties yielding near-zero or negative AAI (de Graaf et al., 2005). We changed the following sentence:

l. 310: "The AAI generally increases in the presence of absorbing aerosols and can, unlike the aerosol optical depth, also be computed in the presence of clouds." -> "The AAI generally increases in the presence of absorbing aerosols and can, unlike the aerosol optical depth, also be computed when the aerosol layer is above clouds."

We chose the AAI to apply the correction to because the signature of a solar eclipse in the AAI is significant in value and spatial size. The AAI application is relatively straightforward because the AAI depends on the absolute reflectances at only the two wavelengths 340 and 380 nm (rather than differential features on a fitted polynomial) and therefore is directly affected by the eclipse obscuration. In a future study, the correction could be tested on products that are derived from the reflectance at more than two wavelengths. We added the following sentence to the Introduction:

l.33: "The AAI is retrieved from TOA reflectance measurements at two wavelengths in the UV-range, hence the AAI may directly be affected by the obscuration during a solar eclipse."

- *Figure 15: The scatter is rather large. So the figure's value to validate the correction method is rather limited for X > 0.7. This could be mentioned.*

Although we have carefully selected the reflectances in orbit 11403 and 11404 to calculate the obscuration factors, the selected reflectances in the orbit 11403 cannot be the same as the reflectances in orbit 11404, due to natural variation across the Earth's surface. There are indeed some points that are scattered, however, most points are located around the calculated results (represented by the opacity of the points in the plot). Making the filter of Equation 15 more strict (e.g. $R_{340}^{\text{meas}} > 0.75 \cdot R_{380}^{\text{meas}}$), decreases the scatter but also decreases the number of points. The scatter is less for small $X$, because the number of points roughly decreases with decreasing $X$ simply because $X$ is directly related to the circumference of the surface area centered at the shadow axis (where $X = 0$) for which the solar to lunar disk center separation is smaller than $X$. We have added a footnote to line 291:

The density of points increases with increasing $X$ because the Earth's surface area for which a certain value of $X$ applies increases with increasing $X$. Making the filter of Equation 15 more strict (e.g. $R_{340}^{\text{meas}} > 0.75 \cdot R_{380}^{\text{meas}}$), decreases the scatter but also decreases the number of points.

*Technical comments*

- L41: "have been taken," > "have been observed,"

We do not agree. The measurements have been taken.

- L51: Why "instead"? The verb "approximated" already implies that this is a simplification.

Montornès et al. (2016) used a different approximation than mentioned in the former sentence. We removed the word "instead", and changed the font type of the word "diameter" to italic:

L51: "The eclipse obscuration at a point in the shadow can be approximated by the fraction of the area of the apparent solar disk occulted by the Moon (Seidelmann, 1992). Montornès et al. (2016) instead approximated the eclipse obscuration by the fraction of the solar disk diameter occulted by the Moon ..." -> "The eclipse obscuration at a point in the shadow can be approximated by the fraction of the area of the apparent solar disk occulted by the Moon (Seidelmann, 1992). Montornès et al. (2016) approximated the eclipse obscuration by the fraction of the solar disk *diameter* occulted by the Moon ..."

– L168: low > small

We changed the text as suggested:

L168: "... a relatively low $r_m$, ..." -> "... a relatively small $r_m$, ..."

– L241: For clarity, please explain "scanline". (E.g., the line at Earth's surface defined by the satellite swath that is roughly oriented East-West)

We have added a footnote after the word "scanline":

L241: The line at the Earth's surface perpendicular to the flight direction defined by the satellite swath which is roughly oriented West-East.

– L275: "Sect.." > "section."

According to https://www.atmospheric-chemistry-and-physics.net/submission.html#templates": "The abbreviation "Sect." should be used when it appears in running text and should be followed by a number unless it comes at the beginning of a sentence." Here, indeed, "Sect." is not followed by a number, because the sentence reads " in this Sect.." Therefore, we think the reviewer correctly states that "Sect." should be written as "section". We thank the reviewer for pointing this out and changed the sentence:

L275: "... in this Sect.." -> "... in this section."

Similarly, we changed:

L403: "In this Sect., we reflect back on the assumptions ..." -> "In this section, we reflect back on the assumptions ..."

– L317 Moon > Moon's

We think that both Moon's shadow and Moon shadow are correct. To be consistent with the the terminology in rest of the paper, we leave term "Moon shadow" here.

– L331: 110405 > 11405

We changed the sentence as suggested. We thank the reviewer for pointing out this typo.

"... orbits 11403 and 110405 ..." -> "... orbits 11403 and 11405..."

- L358: delete "still"

  We changed the sentence as suggested:

  L358: "... still in too low ..." -> "... in too low ..."

245  − L415: What is "chord"?

  We changed "chord" into "straight line":

  L415: "... at the chord ... " -> "... at the straight line ..."

The following appendix is added to the paper:

**Appendix B: Error propagation**

250  In this appendix, we show the effect of the solar irradiance correction on the precision of the AAI. It should be recalled from Equation 3 that the corrected measured TOA reflectance, $R^{\text{int}}(\lambda)$, is computed from the measured TOA reflectance by TROPOMI, $R^{\text{meas}}(\lambda)$, and the calculated obscuration fraction, $f_{\text{o}}(\lambda)$. We assume that the noise of $R^{\text{meas}}(\lambda)$ and the noise of $f_{\text{o}}(\lambda)$ are normally distributed with standard deviations $\sigma_{R^{\text{meas}}(\lambda)}$ and $\sigma_{f_o(\lambda)}$, respectively. Also, we assume that the noise of $R^{\text{meas}}(\lambda)$ is not correlated with the noise of $f_{\text{o}}(\lambda)$. Then, we may compute the precision of $R^{\text{int}}(\lambda)$ as follows

255  $$\sigma_{R^{\text{int}}} = R^{\text{int}} \cdot \sqrt{\left(\frac{\sigma_{R^{\text{meas}}}}{R^{\text{meas}}}\right)^2 + \left(\frac{\sigma_{f_{\text{o}}}}{1 - f_{\text{o}}}\right)^2} \qquad \text{(B1)}$$

$\sigma_{R^{\text{meas}}(\lambda)}$ is provided in the current operational TROPOMI L2 AAI product for $\lambda = 340$ nm and $\lambda = 380$ nm. $\sigma_{f_o(\lambda)}$ depends on the precision of the geometrical eclipse prediction, i.e. $\alpha$ in Equation 9, and the precision of the solar limb darkening function, $\Gamma$. The geometrical eclipse prediction was verified with the predictions by NASA (see Section 2.4). The largest source of uncertainty for $\alpha$ in the present era (1800 CE to present) is the Moon's surface topography, which causes the lunar disk

260  circumference to deviate from a perfect circle.[2] Our and NASA's eclipse predictions do not include these effects of mountains and valleys along the edge of the Moon, which may shift the limits of the eclipse path north or south by $\sim 1$ to $3$ kilometers, and may change the eclipse duration by $\sim 1$ to $3$ seconds.[3] For the solar eclipse of 21 June 2020, we added a time increment of 3 seconds to estimate the effect of a local eclipse timing error due to the Moon's topography on $f_{\text{o}}$ and the AAI. At the ground pixels for which $f_{\text{o}} > 0$, the average absolute changes in $f_{\text{o}}$ and the AAI were 0.00049 and 0.00588, respectively. Hence, in

[revised manuscript text omitted]

Sun, J., Veefkind, P., Nanda, S., van Velthoven, P., and Levelt, P.: The role of aerosol layer height in quantifying aerosol absorption

300    from ultraviolet satellite observations, Atmospheric Measurement Techniques, 12, 6319–6340, https://doi.org/10.5194/amt-12-6319-2019, https://amt.copernicus.org/articles/12/6319/2019/, 2019.

---

## Author Comment (AC2) · 27 Mar 2021

**Response to comment of Anonymous Referee #2 on "Restoring the top-of-atmosphere reflectance during solar eclipses: a proof of concept with the UV Absorbing Aerosol Index measured by TROPOMI" by Victor Trees et al.**

Victor Trees[1,2], Ping Wang[1], and Piet Stammes[1]

[1]Royal Netherlands Meteorological Institute (KNMI), De Bilt, The Netherlands
[2]Delft University of Technology, Delft, The Netherlands

**Correspondence:** Victor Trees (victor.trees@knmi.nl)

We thank the reviewer for his/her careful reading and for the comments and suggestions, which have improved the manuscript. Below, we give in *black italic* the reviewer's comment, in black our response, and in red the changed text in the manuscript.

*The paper describes a method to correct TROPOMI/S5P observations during solar eclipses. The shadow of the moon reduces*
5  *the incident irradiance. In the derivation of reflectances from this observations the irradiance of non-eclipse conditions is used, therefore these reflectances are wrong and retrieval algorithms using these reflectances yield wrong results. Therefore observations during eclipses are currently not used for further analysis. The observations can be corrected quite easily by using the reduced incident irradiance to derive the reflectance. Consistently with other studies, it is shown that in order to compute the reduced irradiance it is important to take into account the solar limb darkening. The authors derived such a correction method*
10  *and apply it to the derivation of the aerosol absorption index. Using the corrected reflectances they obtain reasonable results also during the eclipse which are consisted with observations derived in non-eclipse conditions. Satellite based aerosol and trace gas measurements my reveal interesting effects of the solar eclipse on the composition of the atmosphere, however this is not investigated in the study. The paper is generally well written in good English and the number of figures is appropriate. I recommend publication in ACP after some revisions as suggested in my comments below.*

15

*General comments:*

– *In the paper the method to correct observations during solar eclipses is described. It is mentioned in the introduction that corrected observations can be used to study effects of the solar eclipse on atmospheric composition. I suggest to include such a study, this would increase the scientific relevance of the paper significantly.*

20  We agree with the reviewer that the application of the reflectance correction to trace gas products is of scientific relevance. The goal (and title) of this paper, however, is to show that the TOA reflectance during solar eclipses can be restored, which is proven with the application to the AAI product. Applying the correction to a trace gas product is another application of the method presented in this paper, and is therefore beyond the scope of this paper.

The AAI was the most straightforward product for such a proof of concept, because the AAI depends on the absolute reflectances at only the two wavelengths 340 and 380 nm. The retrieval of a trace gas product such as the NO2 column is based on the differential features on a fitted polynomial through the UV-VIS reflectance spectrum. We note that the accuracy of such an eclipse corrected retrieval may depend on the accuracy and wavelength resolution of the solar limb darkening measurements. Also, high wavelength resolution solar spectrum features that are not captured by the solar limb darkening measurements have to be taken into account in the retrieval. We leave this challenge for a future study, but added the following sentence to the Discussion:

l.462: "However, high wavelength resolution solar spectrum features that are not captured by the solar limb darkening measurements may have to be taken into account in the retrieval."

– *Motivate, why it is interesting to study solar eclipses and their effects on atmospheric composition. In the abstract it is written that it is "may be of particular interest", this sounds as if the authors do not know themselves whether it is really interesting ...*

Using the restored measurements during solar eclipses, our conceptual understanding of the atmospheric response (e.g. solar eclipse induced absorbing aerosol changes or atmospheric chemistry changes) can be enhanced. In this paper, we did not find an indication of absorbing aerosol changes in the Moon shadow (e.g. that are spatially correlated with the recent eclipse ground track). Restored measurements of a trace gas product can be used to potentially detect such eclipse induced trace gas changes and possibly to verify irradiation responses in atmospheric chemistry models.

We changed the sentence in the third paragraph of the Introduction:

l.42: "Studying the speed and significance of this atmospheric response could contribute to the understanding of the sensitivity of planetary atmospheres to (variations in) their irradiation." -> "Measurements of the speed and significance of this atmospheric response could contribute to the understanding of the sensitivity of planetary atmospheres to (variations in) their solar or stellar illumination and could possibly be used to verify atmospheric chemistry models."

We changed this sentence in the Abstract:

l.6: "... may be of particular interest to users studying solar eclipses and their effect on the Earth's atmosphere." -> "... is of particular interest to users studying the atmospheric response to solar eclipses."

We added this sentence to the Abstract:

l.15: "No indication of local absorbing aerosol changes caused by the eclipses was found."

We added this sentence to the Conclusion:

l.489: "In this paper, we did not find an indication of absorbing aerosol changes in the Moon shadow (e.g. which are spatially correlated with the recent eclipse ground track)."

– *How important is this correction method? How frequently are the observations disturbed by solar eclipses?*

55  Every year, there are about 2 to 3 solar eclipses (with a yearly average of 2.4 solar eclipses)[1]. Since the start of the nominal operational mode of TROPOMI in May 2018, 7 solar eclipses occurred, 6 of which have been measured by TROPOMI. In the paper, in the second paragraph of the Introduction, we mention that, sometimes, such eclipse anomalies propagate into anomalies in temporal average maps without raising an eclipse flag, potentially resulting in false conclusions about the mean aerosol effect in that time period. An example of a *monthly average* AAI map of the GOME-2 satellite instrument

60  that is distorted by a solar eclipse can be found on https://d1qb6yzwaaq4he.cloudfront.net/airpollution/absaai/GOME2B/ monthly/images/2019/GOME-2B_AAI_map_201912.png. The formulae in the Appendix of this paper can also be used to only compute an eclipse flag. Indeed, an eclipse flag computation is not new and the eclipse anomalies could be omitted by raising an eclipse flag and discarding the data, however, the advantage of a correction is that data is not lost and can potentially be used to study solar eclipses from space. In addition, the successful correction shows that the

65  physics of the shadow is understood.

We have added the following sentence to the second paragraph of the Introduction:

l.28: "Since the start of the nominal operational mode of the TROPOMI spectrometer instrument on board the S5P satellite in May 2018, 7 solar eclipses occurred, 6 of which have been measured by TROPOMI."

We have changed the last sentence in the second paragraph of the Introduction:

70  l.37: "Sometimes, such eclipse anomalies propagate into anomalies in temporal average maps without raising an eclipse flag, potentially resulting in false conclusions about the mean aerosol effect in that time period." -> "TROPOMI data contains an eclipse flag indicating the eclipse occurrence at a ground pixel. For satellite instruments that do not contain an eclipse flag, such as the GOME-2 instrument, these eclipse anomalies propagate into anomalies in temporal average maps, potentially resulting in false conclusions about the mean aerosol effect in that time period."

75  Also, we have added the following footnote to the second paragraph of the Introduction:

l.37: An example of a *monthly average* AAI map of the GOME-2 satellite instrument that is distorted by a solar eclipse can be found on https://d1qb6yzwaaq4he.cloudfront.net/airpollution/absaai/GOME2B/monthly/images/2019/GOME-2B_ AAI_map_201912.png, visited on 22 February 2021.

*Specific comments:*

80  – *l.1 "Solar eclipses reduce the measured top-of-atmosphere (TOA) reflectances as derived by Earth observation satellites, because the solar irradiance that is used to compute these reflectances is commonly measured before the start of the eclipse." -> This sentence in the beginning is a little confusing, rephrase? First mention that solar irradiance is reduced in moon shadow. Then mention, that normalized quantity "reflectance" should not be affected when reduced irradiance is used for normalization and write that this is not yet done in the operational processing of the data ...*

85  We thank the reviewer for this suggestion. We changed the sentences, l.1:
* * *
[1] See https://eclipse.gsfc.nasa.gov/SEpubs/5MCSE.html, visited on 26 March 2021.

"Solar eclipses reduce the measured top-of-atmosphere (TOA) reflectances as derived by Earth observation satellites, because the solar irradiance that is used to compute these reflectances is commonly measured before the start of the eclipse. Consequently, air quality products that are derived from these spectra, ..." ->

"During a solar eclipse the solar irradiance reaching the top-of-atmosphere (TOA) is reduced in the Moon shadow. The solar irradiance is commonly measured by Earth observation satellites before the start of the eclipse and not corrected for this reduction, which results in a decrease of the computed TOA reflectances. Consequently, air quality products that are derived from TOA reflectance spectra, ..."

– l.12 "in a maximum Moon shadow signature in the AAI of 6.7 points increase" -> what is a "point"?

The AAI is a dimensionless quantity. We often use often 'points' as a 'placeholder unit' for the AAI because it helps in the communication, similar as 'percent points'.

– l.206: "We provide the recipe for the computation of X ... "-> Have you compared your derivation to that presented in Ockenfuss et al. 2020?

The definition of $X$ in this paper is equivalent to the one used in the paper of Ockenfuß et al. (2020). Ockenfuß et al. (2020) retrieved emphemeris data for the Sun and the Moon from a JPL database (Giorgini et al., 1996), in order to compute $X$. We use the Besselian elements precomputed for each eclipse by NASA in order to compute $X$ based on the work of Meeus (1989), Seidelmann (1992) and Espenak and Meeus (2006), as explained in the paper. We did not compare our computation of $X$ to the one of Ockenfuß et al. (2020), because that would be a comparison of the astrodynamical calculations of Meeus (1989), Seidelmann (1992) and Espenak and Meeus (2006) to the calculations of Giorgini et al. (1996). We verified the ground track of $X = 0$ to the benchmark results published by NASA, as mentioned in the last two sentences of Section 2.

– l.293: "The maximum underestimation of $f_o$ at 380 nm, when using $\Gamma = 1$, was 0.06 at $6.04°N$ latitude and $107.19°E$ longitude." -> What is the maximum underestimation when limb darkening is taken into account ...

The underestimation is with respect to the computed $f_o$ at 380 nm taking into account solar limb darkening. To clarify this, we changed the sentence:

l.287: "The maximum underestimation of $f_o$ at 380 nm, when using $\Gamma = 1$, was 0.06 at $6.04°N$ latitude and $107.19°E$ longitude." -> "The maximum underestimation of $f_o$ at 380 nm when using $\Gamma = 1$, with respect to $f_o$ at 380 nm when solar limb darkening is taken into account, was 0.06 at $6.04°N$ latitude and $107.19°E$ longitude."

– l.318: "The negative mean AAI are partly caused by the scattering of cloud droplets, but also due to a radiometric calibration offset and degradation in the TROPOMI irradiance data ..." -> Please explain: 1. Why is AAI negative for cloud scattering, 2. Why is there a radiometric calibration offset, 3. Why is there a degradation in the TROPOMI irradiance data.

1. The negative AAI due to cloud droplets (scattering aerosols) is explained in Section 3.3 of de Graaf et al. (2005): scattering aerosols generally increase the reflectance measured at 380 nm, resulting in a relative high computed scene albedo $A_s$ (Eq. 17 of our paper). Therefore, the spectrally flat surface contribution in the DAK model is relatively high, resulting in a relatively flat UV model spectrum ($R_{340}^{\text{model}}/R_{380}^{\text{model}} \approx 1$). This model spectrum may even be more flat than the measured spectrum, which particularly happens in scenes with intermediate effective cloud fractions (i.e. intermediate geometrical cloud fractions or thin clouds, see Penning de Vries et al. (2009)). Hence, $R_{340}^{\text{meas}}/R_{380}^{\text{meas}} < R_{340}^{\text{model}}/R_{380}^{\text{model}}$ results in a negative AAI (Eq. 17 of our paper). Because the focus of this paper is on the solar eclipse signature, we refer the reader in the first paragraph of Section 3.1.3 to the above mentioned articles for the explanation of the sensitivity of the AAI to scattering by cloud droplets.

2. and 3. During in-flight commissioning some inconsistencies in the on-ground calibration results were found, which concern mainly the absolute irradiance radiometric calibration. In addition, the TROPOMI instrument properties change over time, such as the diffuser degradation or the gain drift of the CCD detector output nodes. More details about the offset and degradation, and the corrections that are applied for the planned version 2 of the L1b processor, can be found in Ludewig et al. (2020).

We have added the reference to Ludewig et al. (2020) to line 328:

l.328: Ludewig et al., (2020)

– *Fig.16: I have a general question about the interpretation AAI. It seems that in the figure most higher values of AAI are not due to aerosols but due to clouds and sunglint? Also values seem to be higher towards the edges of the orbit, are these AAI values correct? Can you indicate an area in the figure which clearly shows an increased AAI due to the presence of aerosol?*

Figure 16 does not contain regions where absorbing aerosols can clearly be identified. Indeed, the most significant AAI features in Fig. 16 are due to the sunglint and clouds (see Kooreman et al., 2020, for the effect of the glint and large-scale clouds on the AAI). An example of an AAI map in which absorbing aerosols are detectable is shown in Figure 1, which we discuss in the Introduction. We have added the following sentence:

l.336: "We note that no significant absorbing aerosol events can be identified in Figure 16."

The AAI indeed slightly increases toward the eastern edges of the orbit swaths at some latitudes. Here, the viewing zenith angles are relatively large. The AAI is based on a comparison of the measured UV reflectances to the UV reflectances modeled by the radiative transfer model DAK, the latter ones being an approximation of reality. de Graaf et al. (2005) showed that the AAI and its accuracy depend on the zenith angles when an absorbing aerosol layer is present (see their Figure 1). An additional possible reason for the AAI increases at some parts of the edges may be the presence of clouds. Although clouds generally decrease or neutralize the AAI (see previous comment), Kooreman et al. (2020) show that anisotropic scattering of light by clouds may increase the AAI at large viewing zenith angles and at the viewing zenith

angle where the cloud bow occurs. These effects, however, also depend on the solar zenith angle and relative azimuth
150   angle of light scattered by the cloud droplets. We refer to Kooreman et al. (2020) for more details.

- *l.385: "at 36°-42°N latitude and 78°-86°E longitude" -> could you mark this region in Fig.20?*

    We have marked the region in Fig. 20 and in Fig. 21 of the paper as shown in Figures 1 and 2 of this document, respectively.

- *l.440: "Hence, the solar irradiance correction of this paper could be used to potentially prove that the yellow and orange*
155   *colors in satellite images are indeed caused by solar limb darkening." -> Can you try this and include a corrected image? This should not be much work?*

    It would be interesting to prove that the yellow and orange colors in VIIRS and/or MODIS true color satellite images are caused by solar limb darkening (TROPOMI does not measure at the green wavelengths and therefore has no true color product). This activity, however, is another application of the eclipse corrected reflectances and therefore beyond
160   the scope of this paper. Indeed, in principle, such an application could be straightforward, but would require a proper implementation and discussion of the reflectance correction in the true color algorithms of VIIRS and/or MODIS, having their own post-proccesing steps that should carefully be considered (for example, the true color images found on https: //worldview.earthdata.nasa.gov are actually corrected for Rayleigh scattering). In order for the paper not to loose focus, we leave this application for a future study.

165   *Technical corrections:*

- *l.17: "can be used to detect real AAI rising phenomena ... " -> "can be used to detect real AAI rising phenomena during a solar eclipse ... "*

    We have changed the text as suggested. l.17: "... can be used to detect real AAI rising phenomena ..." -> "... can be used to detect real AAI rising phenomena during a solar eclipse ..."

170   - *l.90: "and on phi-phi0" -> mention that if 3D effects matter the absolute azimuth angles need to be taken into account*

    Indeed, in case of azimuthally asymmetric reflecting objects such as 3D cloud structures, the reflectance depends on the absolute azimuth angles. We generalized the sentence such that it applies to all types of reflectors. l.92: "... and on $\varphi_0 - \varphi$ which is the viewing azimuth angle relative to the solar azimuth angle." -> ...", on the viewing azimuth angle $\varphi$ and on the solar azimuth angle $\varphi_0$."

[Figure]

**Figure 1.** The Absorbing Aerosol Index from the 340/380 nm wavelength pair by TROPOMI on 21 June 2020 over Asia in orbits 13929-13931, uncorrected (left) and after the solar irradiance correction (right). In the corrected image, the Taklamakan desert is located in the rectangular dotted box.

[Figure]

**Figure 2.** The Absorbing Aerosol Index from the 338/381 nm wavelength pair by GOME-2C on 21 June 2020 over Asia in orbits 8411-8415. The Taklamakan desert is located in the rectangular dotted box.

**References**

de Graaf, M., Stammes, P., Torres, O., and Koelemeijer, R. B. A.: Absorbing Aerosol Index: Sensitivity analysis, application to GOME and comparison with TOMS, Journal of Geophysical Research (Atmospheres), 110, D01201, https://doi.org/10.1029/2004JD005178, 2005.

Espenak, F. and Meeus, J.: Five millennium canon of solar eclipses : -1999 to +3000 (2000 BCE to 3000 CE), NASA Technical Publication TP-2006-214141, 2006.

Giorgini, J. D., Yeomans, D. K., Chamberlin, A. B., Chodas, P. W., Jacobson, R. A., Keesey, M. S., Lieske, J. H., Ostro, S. J., Standish, E. M., and Wimberly, R. N.: JPL's On-Line Solar System Data Service, in: AAS/Division for Planetary Sciences Meeting Abstracts #28, vol. 28 of *AAS/Division for Planetary Sciences Meeting Abstracts*, p. 25.04, 1996.

Kooreman, M. L., Stammes, P., Trees, V., Sneep, M., Tilstra, L. G., de Graaf, M., Stein Zweers, D. C., Wang, P., Tuinder, O. N. E., and Veefkind, J. P.: Effects of clouds on the UV Absorbing Aerosol Index from TROPOMI, Atmospheric Measurement Techniques Discussions, 2020, 1–31, https://doi.org/10.5194/amt-2020-112, https://amt.copernicus.org/preprints/amt-2020-112/, 2020.

Ludewig, A., Kleipool, Q., Bartstra, R., Landzaat, R., Leloux, J., Loots, E., Meijering, P., van der Plas, E., Rozemeijer, N., Vonk, F., and Veefkind, P.: In-flight calibration results of the TROPOMI payload on board the Sentinel-5 Precursor satellite, Atmospheric Measurement Techniques, 13, 3561–3580, https://doi.org/https://doi.org/10.5194/amt-13-3561-2020, https://amt.copernicus.org/articles/13/3561/2020/, 2020.

Meeus, J.: Elements of solar eclipses, 1951-2200, Willman-Bell Inc., Virginia, 1989.

Ockenfuß, P., Emde, C., Mayer, B., and Bernhard, G.: Accurate 3-D radiative transfer simulation of spectral solar irradiance during the total solar eclipse of 21 August 2017, Atmospheric Chemistry & Physics, 20, 1961–1976, https://doi.org/10.5194/acp-20-1961-2020, 2020.

Penning de Vries, M. J. M., Beirle, S., and Wagner, T.: UV Aerosol Indices from SCIAMACHY: introducing the SCattering Index (SCI), Atmospheric Chemistry and Physics, 9, 9555–9567, https://doi.org/10.5194/acp-9-9555-2009, https://acp.copernicus.org/articles/9/9555/2009/, 2009.

Seidelmann, P. K.: Explanatory Supplement to the Astronomical Almanac, U.S. Naval Observatory, Washington, D.C., University Science Books, California, 1992.

---

## Author Response (AR2)

**Response to comment of the editor on "Restoring the top-of-atmosphere reflectance during solar eclipses: a proof of concept with the UV Absorbing Aerosol Index measured by TROPOMI" by Victor Trees et al.**

Victor Trees[1,2], Ping Wang[1], and Piet Stammes[1]

[1]Royal Netherlands Meteorological Institute (KNMI), De Bilt, The Netherlands
[2]Delft University of Technology, Delft, The Netherlands

**Correspondence:** Victor Trees (victor.trees@knmi.nl)

We thank the editor for the suggestion to add a reference to Gil et al. (2000). We have added the following sentences to the Introduction, page 2:

Unambiguous increases in local $NO_2$ concentration have been measured from the ground during solar eclipses resulting from the reduced photodissociation of $NO_2$ in the stratosphere (see e.g. Gil et al., 2000; Adams et al., 2010). Unlike ozone, $NO_2$ reacts on a timescale of several minutes directly responding to the eclipse obscuration (Herman, 1979; Wuebbles and Chang, 1979). Although similar information could be obtained during sunrise and sunset, Wuebbles and Chang (1979) pointed out that the relatively short time durations of solar eclipses allow for a more clear identification of the major photochemical cycles in the stratosphere, due to the smaller bias from atmospheric transport, mixing and interfering chemical reactions throughout the diurnal cycle.

**References**

Adams, C., McLinden, C. A., Strong, K., and Umlenski, V.: Ozone and NO2 variations measured during the 1 August 2008 solar eclipse above Eureka, Canada with a UV-visible spectrometer, Journal of Geophysical Research: Atmospheres, 115, https://doi.org/https://doi.org/10.1029/2010JD014424, https://agupubs.onlinelibrary.wiley.com/doi/abs/10.1029/2010JD014424, 2010.

15  Gil, M., Puentedura, O., Yela, M., and Cuevas, E.: Behavior of $NO_2$ and $O_3$ columns during the eclipse of February 26, 1998, as measured by visible spectroscopy, Journal of Geophysical Research, 105, 3583–3593, https://doi.org/10.1029/1999JD900973, 2000.

Herman, J. R.: The response of stratospheric constituents to a solar eclipse, sunrise, and sunset, Journal of Geophysical Research: Oceans, 84, 3701–3710, https://doi.org/https://doi.org/10.1029/JC084iC07p03701, https://agupubs.onlinelibrary.wiley.com/doi/abs/10.1029/JC084iC07p03701, 1979.

20  Wuebbles, D. and Chang, J. S.: A theoretical study of stratospheric trace species variations during a solar eclipse, Geophysical Research Letters, 6, 179–182, https://doi.org/https://doi.org/10.1029/GL006i003p00179, https://agupubs.onlinelibrary.wiley.com/doi/abs/10.1029/GL006i003p00179, 1979.